# A hierarchical pathway for assembly of the distal appendages that organize primary cilia

**Tomoharu Kanie[1,2]\*, Beibei Liu[2], Julia F Love[3], Saxton D Fisher[3], Anna-Karin Gustavsson[3,4,5,6,7,8], Peter K Jackson[1]\***

[1]Baxter Laboratory, Department of Microbiology & Immunology and Department of Pathology, Stanford University, Stanford, United States; [2]Department of Cell Biology, University of Oklahoma Health Sciences Center, Oklahoma City, United States; [3]Department of Chemistry, Rice University, Houston, United States; [4]Department of BioSciences, Rice University, Houston, United States; [5]Department of Electrical and Computer Engineering, Rice University, Houston, United States; [6]Smalley-Curl Institute, Rice University, Houston, United States; [7]Center for Nanoscale Imaging Sciences, Rice University, Houston, United States; [8]Department of Cancer Biology, University of Texas MD Anderson Cancer Center, Houston, United States

**\*For correspondence:**
Tomoharu-Kanie@ouhsc.edu
(TK);
pjackson@stanford.edu (PKJ)

**Competing interest:** The authors declare that no competing interests exist.

**Abstract** Distal appendages are ninefold symmetric blade-like structures attached to the distal end of the mother centriole. These structures are critical for the formation of the primary cilium, by regulating at least four critical steps: preciliary vesicle recruitment, recruitment and initiation of intraflagellar transport (IFT), and removal of CP110. While specific proteins that localize to the distal appendages have been identified, how exactly each protein functions to achieve the multiple roles of the distal appendages is poorly understood. Here, we comprehensively analyze known and newly discovered distal appendage proteins (CEP83, SCLT1, CEP164, TTBK2, FBF1, CEP89, KIZ, ANKRD26, PIDD1, LRRC45, NCS1, CEP15) for their precise localization, order of recruitment, and their roles in each step of cilia formation. Using CRISPR-Cas9 knockouts, we show that the order of the recruitment of the distal appendage proteins is highly interconnected and a more complex hierarchy. Our analysis highlights two protein modules, CEP83-SCLT1 and CEP164-TTBK2, as critical for structural assembly of distal appendages. Functional assays revealed that CEP89 selectively functions in the RAB34+ vesicle recruitment, while deletion of the integral components, CEP83-SCLT1-CEP164-TTBK2, severely compromised all four steps of cilium formation. Collectively, our analyses provide a more comprehensive view of the organization and the function of the distal appendage, paving the way for molecular understanding of ciliary assembly.

## Editor's evaluation

This fundamental study presents the most comprehensive view of the functional organization and requirements for a mother centriole's distal appendage in primary cilia assembly published to date. Crispr-knockouts and super-resolution microscopy analysis of the distal appendage proteins provides compelling evidence to support the claims of the authors. This work will be of high value to cell biologists and biophysicists working on the structure and function of the centrosome as well as human geneticists exploring ciliary pathology.

## Introduction

The primary cilium is an organelle that extends from the cell surface and consists of the ninefold microtubule-based structure (or axoneme) and the ciliary membrane (*Reiter and Leroux, 2017*). With specific membrane proteins (e.g. G-protein coupled receptors) accumulated on its membrane, the cilium serves as a sensor for the extracellular environmental cues (*Reiter and Leroux, 2017*). Biogenesis of the cilium is coupled to the cell cycle, such that the cilium mainly forms during G0/G1 phase of the cell cycle and disassembles prior to mitosis (*Vorobjev and Chentsov Yu, 1982*). In G0/G1 phase, the cilium extends from the mother (or older) centriole, which is distinguished from the daughter (or younger) centriole by its possession of the two centriolar substructures (*Vorobjev and Chentsov Yu, 1982*): distal appendages and subdistal appendages (*Paintrand et al., 1992*). The distal appendages are ninefold symmetrical blade-like structures with each blade attaching to the triplet microtubules at the distal end of the mother centriole (*Anderson, 1972*; *Bowler et al., 2019*; *Paintrand et al., 1992*). Unlike subdistal appendages, which appear to be dispensable for the cilium formation (*Mazo et al., 2016*), the distal appendages play crucial roles in the cilium biogenesis, through their regulation of at least four different molecular steps of the cilium formation (*Graser et al., 2007*; *Schmidt et al., 2012*; *Sillibourne et al., 2013*; *Tanos et al., 2013*): (1) ciliary vesicle recruitment (*Schmidt et al., 2012*; *Sillibourne et al., 2013*), (2) recruitment of intra-flagellar transport (IFT) protein complexes (*Čajánek and Nigg, 2014*; *Goetz et al., 2012*; *Schmidt et al., 2012*), (3) recruitment of CEP19-RABL2 complex (*Dateyama et al., 2019*), which is critical for IFT initiation at the ciliary base (*Kanie et al., 2017*), and (4) removal of CP110 (*Čajánek and Nigg, 2014*; *Goetz et al., 2012*; *Tanos et al., 2013*), which is believed to suppress axonemal microtubule extension (*Spektor et al., 2007*), from the distal end of the mother centriole. However, how the distal appendages modulate these molecular processes is poorly understood.

To date, ten proteins have been shown to localize to the distal appendages: CEP164 (*Graser et al., 2007*), CEP89 (also known as CCDC123) (*Sillibourne et al., 2013*; *Sillibourne et al., 2011*), CEP83 (*Tanos et al., 2013*), SCLT1 (*Tanos et al., 2013*), FBF1 (*Tanos et al., 2013*), TTBK2 (*Čajánek and Nigg, 2014*), INPP5E (*Xu et al., 2016*), LRRC45 (*Kurtulmus et al., 2018*), ANKRD26 (*Bowler et al., 2019*), and PIDD1 (*Burigotto et al., 2021*; *Evans et al., 2021*). These proteins are recruited to the distal appendages in hierarchical order, where CEP83 sits at the top of the hierarchy and recruits SCLT1 and CEP89 (*Tanos et al., 2013*). SCLT1 recruits CEP164 (*Tanos et al., 2013*), ANKRD26 (*Burigotto et al., 2021*; *Evans et al., 2021*), and LRRC45 (*Kurtulmus et al., 2018*). CEP164 recruits TTBK2 (*Čajánek and Nigg, 2014*). How exactly these proteins function to organize the multiple roles of the distal appendage remains to be elucidated.

Here, we identify three more distal appendage proteins (KIZ, NCS1, and CEP15). CEP15 was previously named as C3ORF14, and we renamed it to CEP15 to reflect its function. The last two will be described in an accompanying paper. With this new set of distal appendage proteins, we sought to provide a comprehensive view of the structure, including precise localization, order of recruitment, and functional role of each distal appendage protein.

## Results

### Localization map of the new set of the distal appendage proteins

Recently, two independent studies determined the precise localization of the classical distal appendage proteins (CEP164, CEP83, SCLT1, FBF1, CEP89, TTBK2, and ANKRD26) using Stochastic Optical Reconstruction Microscopy (STORM) (*Bowler et al., 2019*; *Yang et al., 2018*). We first sought to update the localization map with the new set of distal appendage proteins using 3D-structured illumination microscopy (3D-SIM) in retinal pigment epithelial (RPE) cells. While the lateral (xy) resolution of 3D-SIM is inferior to that of STORM, the flexibility of fluorophore selection and sample preparation for multi-color imaging by 3D-SIM (*Valli et al., 2021*) allows us to easily locate target proteins relative to multiple centriolar markers. Using 3D-SIM, we performed three-color imaging to determine the localization of each distal appendage protein relative to CEP170, a marker for the subdistal appendage and the proximal end of the mother centriole (*Sonnen et al., 2012*), as well as the well-characterized distal appendage protein, CEP164, as references (*Figure 1A*; *Figure 1—figure supplement 1A*). Differential localization of each distal appendage protein relative to CEP164 was readily observed in either top (or axial) view or side (or lateral) view (*Figure 1A*; *Figure 1—figure*

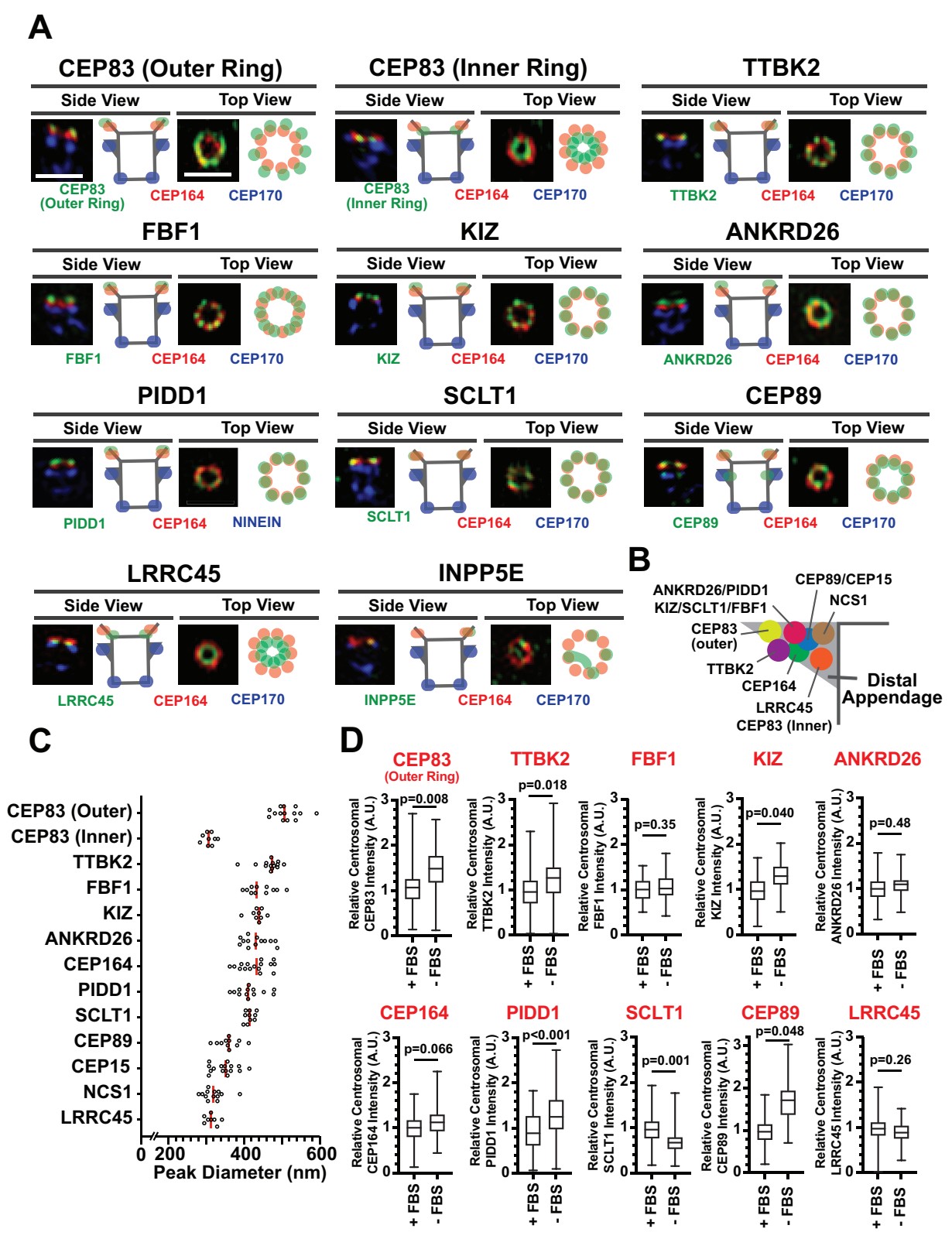

**Figure 1.** Mapping the localization of the distal appendage proteins. (**A**) Retinal pigment epithelial (RPE) cells grown to confluent in FBS-containing media were fixed without serum starvation (for INPP5E), or after the serum starvation for 30 hr (CEP83) or 24 hr (all others). The fixed cells were stained with indicated antibodies and imaged via 3D structured illumination microscopy. Top or Side view pictures of the mother centriole are shown. The individual image is from a representative z-slice. The detailed staining and fixation condition is available in *Figure 1—source data 1*. Scale bar: 1 μm.

*Figure 1 continued on next page*

*Figure 1 continued*

(**B**) The location of each distal appendage protein on the side view of the distal appendage. The model was created from each side view shown in Figure 1A. (**C**) The peak-to-peak diameter of each distal appendage protein. Red bar indicates median diameter. The raw data is available in *Figure 1—source data 2*. (**D**) Quantification of centrosomal signal intensity of indicated distal appendage proteins. RPE cells were grown in FBS-containing media for 24 hr, and then grown in either FBS-containing media or serum free media for additional 24 hr (as shown in *Figure 1—figure supplement 5A*). Cells were fixed and stained with indicated antibodies. Centrosomal signal intensity of each marker was measured from fluorescent image with the method described in materials and methods. The data combined from three independent experiments. Statistical significance was calculated from nested t-test. The raw data, experimental condition, detailed statistics are available in *Figure 1—source data 3*.

The online version of this article includes the following source data and figure supplement(s) for figure 1:

**Source data 1.** Immunofluorescence conditions in the experiment shown in *Figure 1A*.

**Source data 2.** Raw quantification data of the experiment shown in *Figure 1C*.

**Source data 3.** Raw quantification data and detailed statistics of the experiment shown in *Figure 1D*.

**Figure supplement 1.** Individual channels of the images shown in *Figure 1A*.

**Figure supplement 2.** Characterization of the two CEP83 antibodies.

**Figure supplement 2—source data 1.** The original files of the full raw unedited blots shown in *Figure 1—figure supplement 2B*.

**Figure supplement 2—source data 2.** The uncropped blots with boxes that indicate the regions displayed in *Figure 1—figure supplement 2B*.

**Figure supplement 3.** Structural model of CEP83.

**Figure supplement 4.** Localization of N-terminally and C-terminally GFP tagged CEP83.

**Figure supplement 4—source data 1.** Raw quantification data of the experiments shown in *Figure 1—figure supplement 4B,D,F and H*.

**Figure supplement 5.** Confirmation of experimental appropriateness of the data shown in *Figure 1D*.

**Figure supplement 5—source data 1.** Raw quantification data, immunofluorescence conditions and detailed statistics of the experiment shown in *Figure 1—figure supplement 5B*.

**Figure supplement 5—source data 2.** Raw quantification data and immunofluorescence conditions of the experiment shown in *Figure 1—figure supplement 5C*.

---

*supplement 1A*). As an example, the localization of FBF1, which was positioned between adjacent CEP164 structures seen in the axial view of the published STORM picture (which the authors thus identified FBF1 as a distal appendage matrix protein) (*Yang et al., 2018*), was also recapitulated in our SIM image (see FBF1 top view in *Figure 1A*; *Figure 1—figure supplement 1A*). We also observed localization of CEP89 near the subdistal appendage in addition to its distal appendage localization (CEP89 side view in *Figure 1A*), consistent with the previous report (*Chong et al., 2020*; *Yang et al., 2018*). Although localization of the classical distal appendage proteins was essentially the same as that shown by two-color direct STORM imaging (*Yang et al., 2018*), there was one notable difference. CEP83 was shown to localize to the innermost position of the distal appendage (*Bowler et al., 2019*; *Yang et al., 2018*). We recapitulated this localization (see CEP83 inner ring in *Figure 1A*; *Figure 1B*) with the antibody used in the previous two papers (*Bowler et al., 2019*; *Yang et al., 2018*), which recognizes the C-terminal region of CEP83 (*Figure 1—figure supplement 2A*). The peak-to-peak diameter of the inner CEP83 ring (308.6±4.9 nm, *Figure 1C*) was comparable to the previous report (313±20 nm) (*Yang et al., 2018*). However, when we detected this protein with the antibody that detects the middle part of the protein (*Figure 1—figure supplement 2A*), we observed the ring located at the outermost part of the distal appendage with the diameter of 513.4±9.0 nm (see CEP83 outer ring in *Figure 1A*; *Figure 1B*; *Figure 1C*). Both antibodies recognized the specific band at 80–110 kDa that is lost in the *CEP83* knockout cells (*Figure 1—figure supplement 2B*). One intriguing explanation for the difference in the ring diameter is that the protein might have an extended configuration that spans 100 nm (the difference in radius between inner and outer ring) in length. Human CEP83 protein (Q9Y592) is predicted to contain two conserved coiled-coil domains, one of which may span between 40–633 amino acids. Examination of the AlphaFold model (*Jumper et al., 2021*) of CEP83 shows a consistent highly extended alpha-helical structure, conserved among most species (*Figure 1—figure supplement 3*), supporting the model of CEP83 as a highly extended molecule. Since one alpha helix contains 3.6 residues with a distance of 0.15 nm per amino acid (*Pauling et al., 1951*) and contour length of an amino acid is around 0.4 nm (*Ainavarapu et al., 2007*), 400 amino acids stretch of coiled-coil domain as well as a disordered region that consists of 40 amino acids (the maximum distance between antigens of the two CEP83 antibodies) could

contribute at least 76 nm. Since the length of immunoglobulin G (IgG) is ~8 nm (see Figure 2 of *Tan et al., 2008*), the primary and secondary antibodies at the two edges could contribute another ~30 nm. The summed contribution from CEP83 and antibodies could readily account for the ~100 nm difference in radius between the two rings, deriving from the extended structure of CEP83. This model, wherein CEP83 forms an extended backbone scaffolding the distal appendage, is attractive given that CEP83 is important for localization of all the other distal appendage proteins (*Tanos et al., 2013*) to different positions along the distal appendages (*Figure 1B*). Another explanation is that the two antibodies might detect different CEP83 isoforms. We think this is less likely because we detect a single band that is lost in *CEP83* knockouts using the two different antibodies (*Figure 1—figure supplement 2B*). We could also not identify different isoforms of CEP83 with similar size in the Uniprot protein database. To further support our hypothesis, we tagged Green Fluorescent Protein (GFP) to either the N-terminus or the C-terminus of CEP83, and compared the relative position to CEP164 as well as the diameter of the two proteins. Consistent with the endogenous CEP83 detected with the antibody that recognizes C-terminus of the protein, GFP fused to the C-terminus of CEP83 (CEP83-GFP) localized to more inward and more basal centriolar position than CEP164 with the peak-to-peak diameter of 339.7±18.4 nm, when the GFP was detected by antibody (*Figure 1—figure supplement 4A–B*). GFP attached to the N-terminus of CEP83 (GFP-CEP83) was located at the similar position to CEP164 with the diameter of 425.5±17.5 nm (*Figure 1—figure supplement 4C–D*). Note that GFP-CEP83 ring was smaller than the outermost ring of endogenous CEP83, possibly because the difference between N-terminus and the region detected by antibody (somewhere between a.a. 226–568 of the CEP83 isoform 2). We observed the same trend when we detected the tagged proteins with native GFP fluorescent signal instead of antibody, while the difference between GFP-CEP83 and CEP83-GFP was less pronounced possibly due to lower fluorescent signal, which could alter image fidelity (*Demmerle et al., 2017*) as well as inaccurate measurement (*Figure 1—figure supplement 4E–H*). Nonetheless, these data support the model wherein CEP83 forms an extended structure that spans the innermost to the outermost region of the distal appendages to serve as a molecular scaffold. For the recently identified distal appendage protein, PIDD1 (*Burigotto et al., 2021*; *Evans et al., 2021*), we observe a ring with a similar diameter to CEP164 but that was displaced distally in the side view (see PIDD1 in *Figure 1A*; *Figure 1B*). This localization is consistent with the localization of its functional partner ANKRD26 (*Burigotto et al., 2021*; *Evans et al., 2021*; *Figure 1A*; *Figure 1B*). KIZ (or Kizuna) was localized to the similar position to ANKRD26-PIDD1. LRRC45 was located at the innermost region of the distal appendage with the smallest ring diameter (*Figure 1A*; *Figure 1B*; *Figure 1C*). This localization is similar to that of the inner ring of CEP83. INPP5E was reported to localize to the distal appendage in the cells grown with serum and redistribute to the cilium once cells form the organelle upon serum starvation (*Xu et al., 2016*). This localization is also supported by the physical interaction between INPP5E and the distal appendage protein, CEP164 (*Humbert et al., 2012*). Indeed, we observed the INPP5E signal around the distal appendage protein CEP164 in cells grown with serum, however, we rarely observed a ninefold ring (INPP5E in *Figure 1A*), typically observed with all other distal appendage proteins. Therefore, we were unable to measure the diameter of the ring formed by INPP5E. From this result, we think INPP5E is at least not a stable component of distal appendages and instead transiently localizes around the distal appendage. This localization pattern is similar to what was observed for ARL13B (see Figure 4A of *Yang et al., 2018*). Localization of the novel distal appendage proteins, NCS1 and CEP15, are described in an accompanying paper (*Kanie et al., 2025*), but the predicted location and their diameter are shown here for convenience (*Figure 1B*; *Figure 1C*).

Given that cilium formation of RPE cells is induced by serum deprivation (*Figure 1—figure supplement 5A–B*), we next tested if the localization of the distal appendage proteins changes during ciliogenesis (method described in *Figure 1—figure supplement 5A*). Consistent with its function in ciliogenesis, we observed enhanced centriolar localization of IFT88, which requires CEP164 and TTBK2 (*Goetz et al., 2012*; *Schmidt et al., 2012*), upon serum removal (*Figure 1—figure supplement 5C*). Localization of one set of the distal appendage proteins (outer ring of CEP83, TTBK2, KIZ, PIDD1, and CEP89) were significantly enhanced following the serum starvation, whereas another set of distal appendage proteins (FBF1, ANKRD26, CEP164, and LRRC45) were not affected (*Figure 1D*). SCLT1 was the only protein that decreased its centrosomal signal upon serum deprivation.

## Updating the hierarchical map of distal appendage proteins

Distal appendage proteins are recruited to their precise location in a hierarchical order. The previous work described the order of recruitment with a simple epistatic organization, where CEP83 recruits CEP89 and SCLT1, which in turn recruits CEP164 and FBF1 (*Tanos et al., 2013*). With the updated set of distal appendage proteins, we sought to refine the hierarchical map of the distal appendage proteins. Notably, the original epistasis pathway using siRNA knockdowns for loss of function may fail to identify some strong requirements if limited amounts of protein are sufficient for pathway function. To this end, we generated CRISPR-Cas9-mediated knockout cells for each distal appendage protein (*Table 1A and B*). We then tested the localization of each distal appendage protein in each knockout cells via immunofluorescence microscopy. We combined conventional micrographs with semi-automated measurement of centrosomal signal intensity to more accurately quantify the loss of localization of the proteins (see Materials and Methods). In most cases, the centrosomal signal of distal appendage proteins were barely or not detected in their respective knockout cells (see for example *Figure 2A, G, I*), confirming that the antibodies detect specific proteins and that the semi-automated intensity measurement was working properly. In some cases, signal was detected even in the respective knockout cells, because of the high signal observed outside of the centrosome (see for example *Figure 2*), which likely results from non-specific staining of the antibodies. Centriolar FBF1 signal looked specific as it is almost completely lost in *CEP83* knockout cells, however, a weak signal of FBF1 was detected in *FBF1* knockout cells (*Figure 2E*). Since both two *FBF1* knockout clones had one nucleotide insertion in both alleles between coding DNA 151 and 152 (151_152insT) (*Table 1B*), which results in a premature stop codon at the codon 61, we assumed that the knockout cells express truncated protein via either alternative translation or alternative splicing. We could not confirm the truncated protein because of the lack of an antibody that works well for immunoblotting, and currently do not know the functional significance of the truncated proteins. Nonetheless, our semi-automated workflow provides objective and quantitative data to generate an accurate hierarchical map of the distal appendage proteins. Consistent with the original study mapping new distal appendage proteins (*Tanos et al., 2013*), the centrosomal signal intensity of all the knockout cells were greatly diminished in *CEP83* knockout cells (*Figure 2A–K*; *Figure 2M*), confirming that CEP83 is the most upstream. SCLT1 depletion also showed substantial loss of localization of all the other distal appendage proteins, including CEP83 and CEP89 (*Figure 2B–C*; *Figure 2F*), suggesting that CEP83 and SCLT1 organize the distal appendage structure in a co-dependent manner. This is consistent with the previous electron micrograph showing the absence of visible distal appendages in *SCLT1$^{-/-}$* cells (Figure 7B of *Yang et al., 2018*). Note that the *SCLT1* knockout affected the localization of the outer ring of CEP83 much more strongly than the inner ring population (*Figure 2B*; *Figure 2C*). We observed a much stronger effect of *SCLT1* knockouts than the previous report (*Tanos et al., 2013*), which used siRNA to deplete SCLT1, likely because of the complete absence of the protein in our knockout system. Consistent with the previous report (*Čajánek and Nigg, 2014*), TTBK2 localization at the distal appendage was largely dependent on CEP164, as the TTBK2 signal in *CEP164* knockout cells was at an undetectable level similar to *TTBK2* knockout cells (*Figure 2H*). Downstream of CEP164, *TTBK2* knockout affected localization of CEP164, the outer ring CEP83, FBF1, ANKRD26, PIDD1, and NCS1 (*Figure 2A*; *Figure 2C*; *Figure 2E*; *Figure 2G*; *Figure 2K*; *Figure 2M*). Loss of NCS1 localization in *TTBK2* knockout cells is addressed in *Kanie et al., 2025* an accompanying paper (see Figure 2—figure supplement 4D of *Kanie et al., 2025* an accompanying paper). The decrease of CEP164 intensity in *TTBK2* knockout cells is consistent with the observation that CEP164 is a substrate of TTBK2 and that centriolar CEP164 localization was markedly increased upon overexpression of wild type but not kinase-dead TTBK2 (*Čajánek and Nigg, 2014*). Interestingly, localization of another TTBK2 substrate, CEP83 (*Bernatik et al., 2020*; *Lo et al., 2019*), was also affected by TTBK2 depletion (*Figure 2C*). Since only the outer ring of CEP83 was affected by *TTBK2* knockout, we predict that CEP83 phosphorylation by TTBK2 induces a conformational change of CEP83 enabling the protein to organize the backbone of the distal appendage. The localization changes of FBF1, ANKRD26, PIDD1, and NCS1 also suggest that these proteins may be potential substrates of TTBK2, or the localization of these proteins may be affected by phosphorylation of CEP83 or CEP164. In any case, the effect of TTBK2 depletion emphasizes that CEP164-TTBK2 complex (*Čajánek and Nigg, 2014*) organizes a positive feedback loop for other distal appendage proteins to maintain the structural integrity of the distal appendages (*Figure 2N*). Interestingly, ANKRD26 depletion drastically affected not only its

**Table 1.** Guide RNA sequence and genomic DNA analysis of the knockout cells generated and analyzed in this paper.

A

| Cell line | GuideRNA sequence | Direction | Region | Validated by genomic PCR? | Validated by IF? | Validated by Immuno blot? |
|---|---|---|---|---|---|---|
| RPE-BFP-Cas9 CEP89 knockout cells | AATTGGCAGAGTGAGA | Forward | hCEP89 CCDS310-325 | yes | yes | yes |
| RPE-BFP-Cas9 CEP15 knockout cells | GGTGATCAACACACAGAAA | Forward | hC3orf14 CCDS100-118 | yes | no | no |
| RPE-BFP-Cas9 NCS1 knockout cells | TTGTGGAGGAGCTGACC | Forward | hNCS1 CCDS35-51 | no | yes | yes |
| RPE-BFP-Cas9 CEP164 knockout cells | GTTTCCACTCTCCAGGCAG | Reverse | hCEP164 CCDS172-190 | no | yes | yes |
| RPE-BFP-Cas9 CEP83 knockout cells | CTAATTATCAGACACTGA | Forward | hCEP83 CCDS140-157 | yes | yes | yes |
| RPE-BFP-Cas9 SCLT1 knockout cells | TTCCTACCTCTGTGCCCAG | Reverse | hSCLT1 CCDS343-361 | yes | yes | yes |
| RPE-BFP-Cas9 FBF1 knockout cells | TTCTCGCCTTTGAAGAA | Reverse | hFBF1 CCDS105-121 | yes | yes* | no |
| RPE-BFP-Cas9 ANKRD26 knockout cells | GTCCGAGACCGAGATCT | Forward | hANKRD26 CCDS124-140 | no | yes | yes |
| RPE-BFP-Cas9 TTBK2 knockout cells | GAAAATGTTGCACTGAAGG | Forward | hTTBK2 CCDS133-151 | no | yes | no |
| RPE-BFP-Cas9 INPP5E knockout cells | GAAGGGAGGACGCTCCA | Forward | hINPP5E CCDS55-71 | no | yes | no |
| RPE-BFP-Cas9 KIZ knockout cells | GTGCACGAGGGGATTAACTC | Forward | hKIZ CCDS412-431 | yes | yes | yes |
| RPE-BFP-Cas9 LRRC45 knockout cells | ACACCGTGCTGCGCTTTC | Forward | hLRRC45 CCDS254-271 | yes | yes | no |
| RPE-BFP-Cas9 RAB34 knockout cells | GCGGAGGGATCGCGTCCTGG | Forward | hRab34 CCDS21-40 | no | yes | yes |
| RPE-BFP-Cas9 MYO5A knockout cells | GCTACTGCAAAGATATG | Reverse | hMyo5A CCDS412-428 | no | yes | yes |
| RPE-BFP-Cas9 sgIFT52 | AAGAAATATCTTGACAC | Forward | hIFT52 CCDS220-236 | no | no | yes |
| RPE-BFP-Cas9 sgSafe knockout cells | GTCAGTTCCTATGTGGCA | N.A. | N.A. | N.A. | N.A. | N.A. |
| RPE-BFP-Cas9 sgGFP knockout cells | GACCAGGATGGGCACCACCC | Reverse | EGFP CCDS32-51 | N.A. | N.A. | N.A. |

B

| Cell line | Indel in 1st allele | Indel in 2nd allele |
|---|---|---|
| CEP83 knockout_clone6 | 10 bp insertion | 10 bp insertion |
| CEP83 knockout_clone10 | 1 bp insertion | 5 bp deletion |
| SCLT1 knockout_clone1 | 1 bp deletion | 1 bp deletion |
| SCLT1 knockout_clone2 | 1 bp deletion | 1 bp deletion |
| SCLT1 knockout_clone4 | 13 bp deletion | 13 bp deletion |
| SCLT1 knockout_clone6 | 8 bp deletion | 8 bp deletion |
| CEP89 knockout_clone2 | 2 bp deletion | 2 bp insertion |
| CEP89 knockout_clone4 | 5 bp deletion | 5 bp deletion |

*Table 1 continued on next page*

*Table 1 continued*

**A**

| | | |
|---|---|---|
| CEP89 knockout_clone5 | 4 bp deletion | 4 bp deletion |
| CEP89 knockout_clone6 | 155 bp deletion | 117 bp insertion |
| CEP15 knockout_clone5 | 20 bp deletion | 20 bp deletion |
| CEP15 knockout_clone9 | 1 bp insertion | 1 bp insertion |
| CEP15 knockout_clone10 | 145 bp deletion | 145 bp deletion |
| CEP15 knockout_clone13 | 1 bp insertion | 1 bp insertion |
| CEP15 knockout_clone23 | 8 bp deletion | 14 bp deletion |
| LRRC45 knockout_clone1 | 19 bp deletion | 19 bp deletion |
| LRRC45 knockout_clone3 | 11 bp deletion | 11 bp deletion |
| LRRC45 knockout_clone5 | 14 bp deletion | 14 bp deletion |
| LRRC45 knockout_clone6 | 64 bp deletion | 64 bp deletion |
| KIZ knockout_clone1 | 1 bp insertion | 1 bp insertion |
| KIZ knockout_clone2 | 11 bp deletion | 1 bp insertion |
| KIZ knockout_clone12 | 1 bp deletion | 13 bp deletion |
| FBF1 knockout_Clone42 | 1 bp insertion | 1 bp insertion |
| FBF1 knockout_Clone46 | 1 bp insertion | 1 bp insertion |

functional partner, PIDD1 (*Burigotto et al., 2021*; *Evans et al., 2021*; *Figure 2K*), but also the outer ring of CEP83 (*Figure 2C*). This might suggest that ANKRD26 may be crucial for maintaining the protein structure of CEP83 at the outer part of the distal appendages. The diminished CEP164 level in *ANKRD26* knockout cells (*Figure 2A*; *Figure 2M*) might be explained by its direct effect or an indirect effect through the outer ring of CEP83. In contrast to a previous publication (*Kurtulmus et al., 2018*), we did not see the effect of LRRC45 on FBF1 localization (*Figure 2E*). This difference might come from the difference in the experimental setting (e.g. siRNA versus knockout). The signal intensity of a marker of the subdistal appendage, CEP170, was not affected by any of the distal appendage proteins (*Figure 2L*) suggesting that distal appendage proteins are not required for the localization of subdistal appendage proteins at least in terms of signal intensity. The localization changes were mostly not due to the changes in the expression level except that protein KIZ was highly destabilized in *SCLT1* knockout cells (*Figure 2—figure supplement 2A*).

In summary, our updated hierarchical map of distal appendage proteins shows that distal appendage proteins are highly interconnected, and that the organization of the distal appendages is more complex than previously described (*Figure 2N*). The analysis also highlighted the two modules that are critical for maintaining structural integrity of the distal appendage proteins: a CEP83-SCLT1 structural module and a CEP164-TTBK2 module providing a phosphorylation-driven positive feedback module.

## RAB34 is a superior marker for the centriole-associated vesicle

We next sought to understand effector functions for each distal appendage protein. The most well-established function of the distal appendages is recruitment of the preciliary vesicle (*Schmidt et al., 2012*), a precursor for the ciliary membrane, at the early stage of the cilium biogenesis. The recruited ciliary vesicles, called the distal appendage vesicles (*Lu et al., 2015*), then fuse to form a larger ciliary vesicle through mechanisms organized by the Eps15 Homology Domain Protein 1 (EHD1) and Protein kinase C and Casein kinase 2 Substrate in Neurons (PACSIN1 and 2) proteins (*Insinna et al., 2019*; *Lu et al., 2015*). Classically, the method to analyze preciliary vesicle recruitment to the distal appendage was elaborate electron microscopy analysis, largely due to the lack of a preciliary or distal appendage

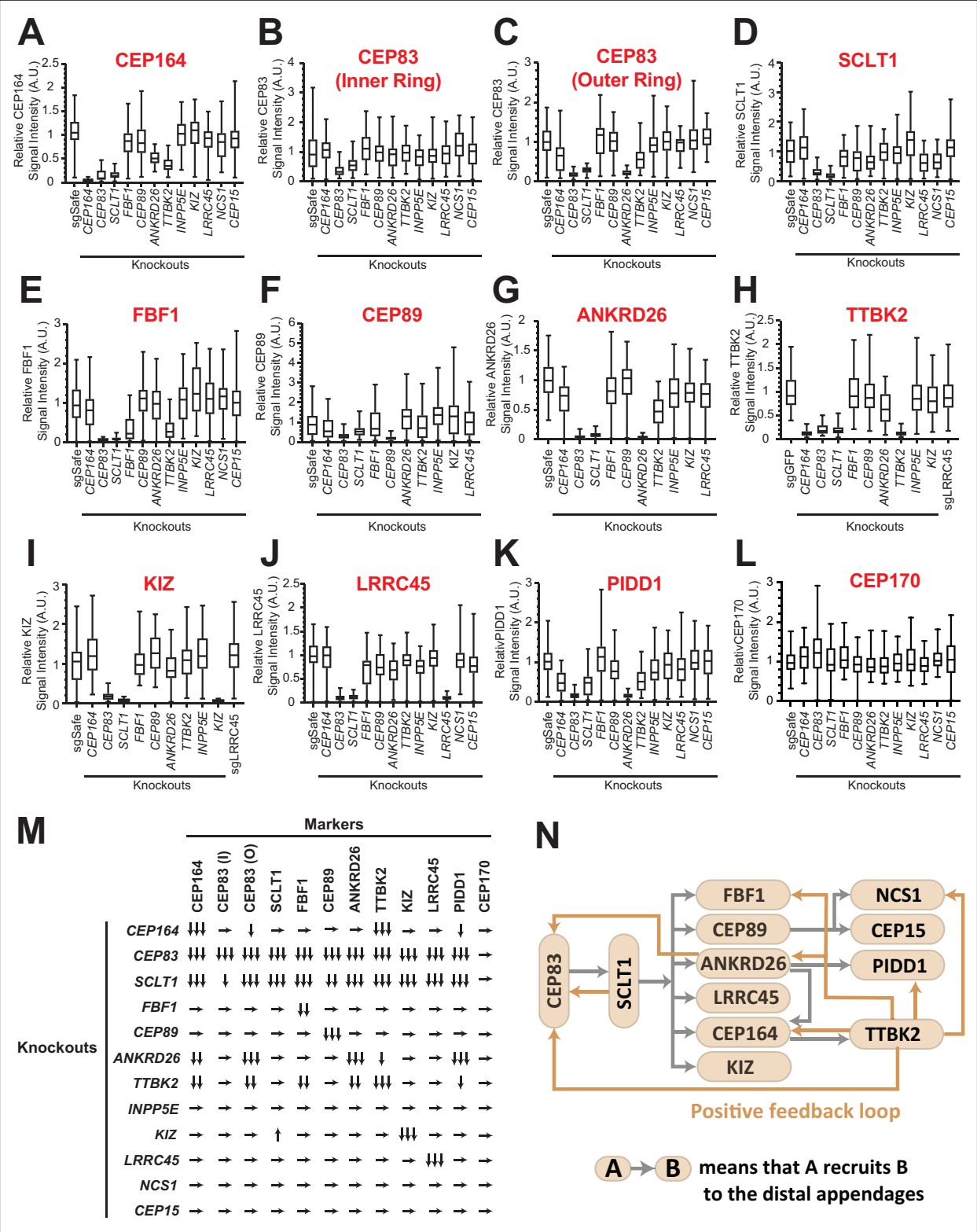

**Figure 2.** The updated hierarchy of the distal appendage proteins. (**A–L**) Box plots showing centrosomal signal intensity of indicated distal appendage proteins (**A–K**) and the subdistal appendage protein, CEP170 (**L**) in retinal pigment epithelial (RPE) cells (control or indicated knockouts) serum-starved for 24 hr (**A, C–L**) or without serum starvation (**B**). The relative fluorescence signal intensity compared with the average of the control is shown. The data from a representative experiment. Note that FBF1 signal remains in *FBF1* knockout cells, and this issue is discussed in the main text. The raw data and experimental condition are available in *Figure 2—source data 1*, *Figure 2—source data 2*, *Figure 2—source data 3*, *Figure 2—source data 4*,

*Figure 2 continued on next page*

Figure 2 continued

*Figure 2—source data 5*, *Figure 2—source data 6*, *Figure 2—source data 7*, *Figure 2—source data 8*, *Figure 2—source data 9*, *Figure 2—source data 10*, *Figure 2—source data 11* and *Figure 2—source data 12*. (**M**) The summary of the signal change in each marker in indicated knockout cells compared with a control. The summary concluded from at least two independent experiments. ↓, weakly reduced; ↓↓, moderately decreased; ↓↓↓, greatly decreased or absent; ↑, weakly increased; →, unaffected. The detailed relationship between CEP89-NCS1-CEP15 as well as localization of each distal appendage protein in *NCS1* knockout cells are available in an accompanying paper (*Kanie et al., 2025*). (**N**) The updated hierarchy of the distal appendage proteins. A→B indicates that A is required for the centrosomal localization of B. CEP83 and SCLT1 are required for each other's localization and are upstream of all the other distal appendage proteins. The outer ring, but not the inner ring, localization of CEP83 was affected by knockouts of several distal appendage proteins (ANKRD26, TTBK2, and CEP164).

The online version of this article includes the following source data and figure supplement(s) for figure 2:

**Source data 1.** Immunofluorescence conditions, and raw quantification data of the experiment shown in *Figure 2A*.

**Source data 2.** Immunofluorescence conditions, and raw quantification data of the experiment shown in *Figure 2B*.

**Source data 3.** Immunofluorescence conditions, and raw quantification data of the experiment shown in *Figure 2C*.

**Source data 4.** Immunofluorescence conditions, and raw quantification data of the experiment shown in *Figure 2D*.

**Source data 5.** Immunofluorescence conditions, and raw quantification data of the experiment shown in *Figure 2E*.

**Source data 6.** Immunofluorescence conditions, and raw quantification data of the experiment shown in *Figure 2F*.

**Source data 7.** Immunofluorescence conditions, and raw quantification data of the experiment shown in *Figure 2G*.

**Source data 8.** Immunofluorescence conditions, and raw quantification data of the experiment shown in *Figure 2H*.

**Source data 9.** Immunofluorescence conditions, and raw quantification data of the experiment shown in *Figure 2I*.

**Source data 10.** Immunofluorescence conditions, and raw quantification data of the experiment shown in *Figure 2J*.

**Source data 11.** Immunofluorescence conditions, and raw quantification data of the experiment shown in *Figure 2K*.

**Source data 12.** Immunofluorescence conditions, and raw quantification data of the experiment shown in *Figure 2L*.

**Figure supplement 1.** Representative immunofluorescence images for the cells analyzed in *Figure 2A–L*.

**Figure supplement 2.** The expression level of distal appendage proteins in the individual distal appendage knockouts.

**Figure supplement 2—source data 1.** The original files of the full raw unedited blots shown in *Figure 2—figure supplement 2A*.

**Figure supplement 2—source data 2.** The uncropped blots with boxes that indicate the regions displayed in *Figure 2—figure supplement 2A*.

vesicle marker. Recently, an unconventional actin-dependent motor protein, Myosin Va (MYO5A), was discovered as the earliest marker for the preciliary/distal appendage vesicle (*Wu et al., 2018*). EHD1 is then recruited to the MYO5A-positive vesicle (*Wu et al., 2018*) to promote fusion and extension of the vesicles. However, MYO5A did not appear to be the best marker for the vesicle, because MYO5A also regulates multiple vesicle trafficking pathways including melanosome transport (*Wu et al., 1998*) and transport of the endoplasmic reticulum (*Wagner et al., 2011*). In agreement with the role of MYO5A outside the cilium, mutations in MYO5A cause Griscelli syndrome (*Pastural et al., 1997*), characterized by hypopigmentation, neurological impairment, and hypotonia, characteristics distinct from other ciliopathies (*Reiter and Leroux, 2017*). The albinism is likely due to the defect in melanosome transport (*Pan et al., 2024*; *Westbroek et al., 2003*) rather than dysfunction of the cilium. When we performed immunofluorescence microscopy, we observed a single punctum of MYO5A that colocalizes with the centrosomal marker CEP170 (*Figure 3A*; arrow in *Figure 3—figure supplement 1A*), consistent with the previous study (*Wu et al., 2018*). We also observe strong MYO5A staining surrounding the centrosome (arrowhead in *Figure 3—figure supplement 1A*). This pericentriolar staining persists in cells deficient in CEP83 (arrowhead in the bottom of *Figure 3—figure supplement 1A*), the structural component of the distal appendages (*Figure 2N*). Because MYO5A shows both centriolar and pericentriolar signals, it is not the best marker for the preciliary/distal appendage vesicles.

To overcome this problem, we searched for other markers for the preciliary/distal appendage vesicle. A recent paper suggested that the small GTPase RAB34, localizes to the centriole-associated vesicle (*Stuck et al., 2021*) and is important for preciliary vesicle recruitment/formation (*Xu et al., 2018*) or for fusion of the distal appendage vesicle (*Ganga et al., 2021*). We first tested whether RAB34 works as the early vesicle marker by staining RPE cells with RAB34, MYO5A, and a centriole marker, CEP170 (*Figure 3A*). We found that MYO5A and RAB34 localization are highly coupled. Most RAB34-positive centrioles have MYO5A at the centriole and vice versa (*Figure 3A*; *Figure 3B*). We confirmed the specificity of the signal using RAB34 (*Figure 3—figure supplement 1B*) and *MYO5A*

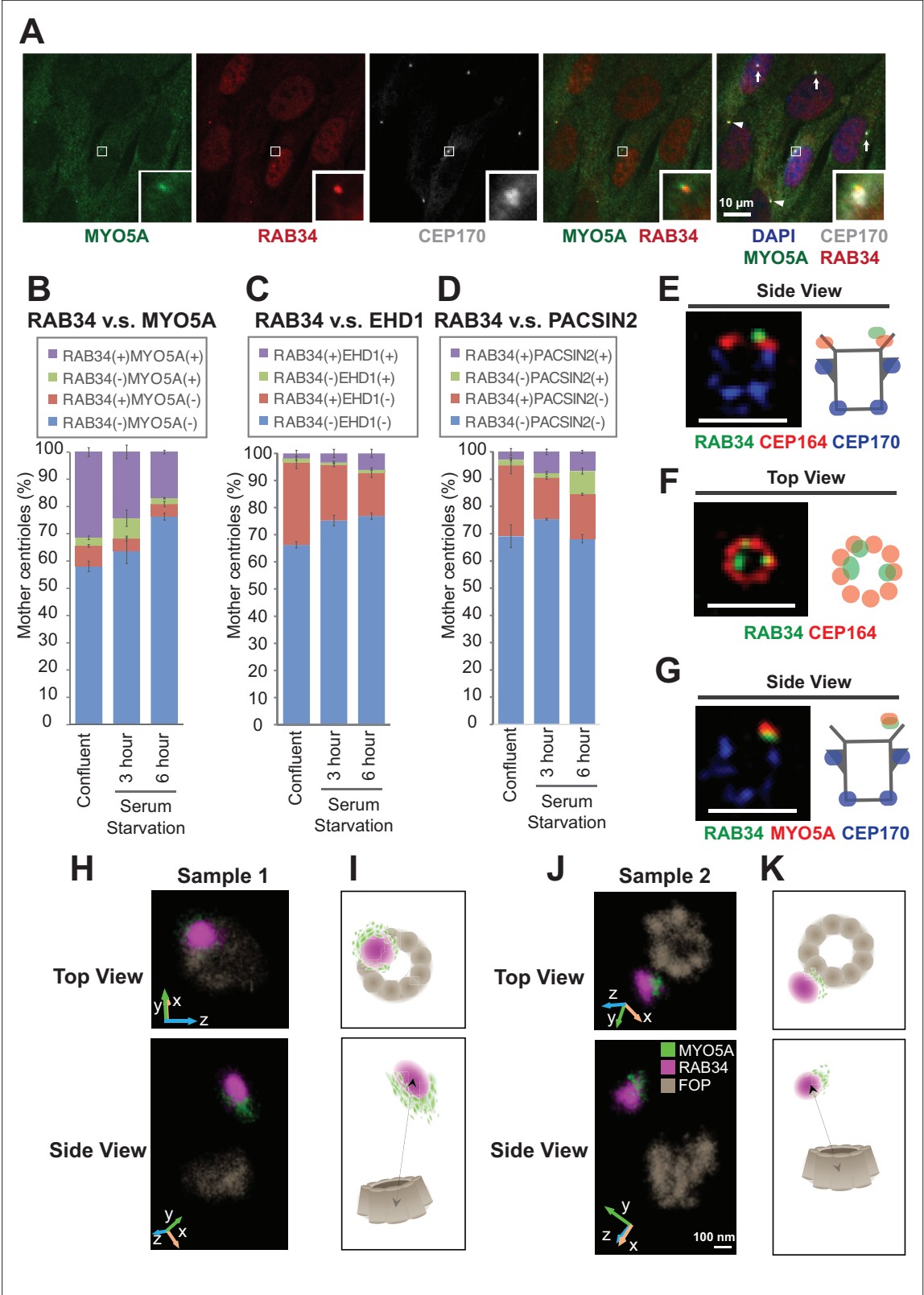

**Figure 3.** RAB34 is a marker for the centriole-associated vesicle. (**A**) Retinal pigment epithelial (RPE) cells were grown in 10% FBS-containing media (serum-fed), fixed, stained with indicated antibodies, and imaged via wide-field microscopy. Arrows and arrowheads indicate RAB34/MYO5A negative or positive centrioles, respectively. Insets at the bottom right corner of each channel are the enlarged images of the smaller insets of each channel. Scale bar: 10 µm. (**B–D**) Quantification of the percentage of the centrioles positive for indicated markers in RPE cells grown in FBS-containing media (**B**) or in

*Figure 3 continued on next page*

*Figure 3 continued*

serum-free media for 3 (**C**) or 6 (**D**) hr. Data are averaged from three experiments. Error bars represent ± SEM. Key statistics are available in *Figure 3—figure supplement 2*. The raw data, sample numbers, experimental conditions, detailed statistics are available in *Figure 3—source data 2*, *Figure 3—source data 3*, and *Figure 3—source data 4*. (**E–G**) RPE cells were grown to confluent in 10% FBS-containing media (serum-fed), fixed, stained with indicated antibodies, and imaged via 3D structured illumination microscopy. Scale bar: 1 μm. (**H–K**) 3D super-resolution reconstructions and illustrations of RAB34 (magenta), MYO5A (green), and FOP (gray). (**H**) and (**J**) Experimental data shown for top and side views relative to the FOP ring-structure. Orientations in the microscope 3D space are indicated by the inset axes. (**I**) and (**K**) Corresponding schematics illustrating the data and highlighting the manner in which MYO5A is located at the edge of the RAB34 distribution. FOP is here visualized with ninefold symmetry. Arrows in the bottom panels indicate measurements of the distance of the RAB34 distribution from the mother centriole FOP structure. The schematics are not drawn to scale. Scale bar: 100 nm.

The online version of this article includes the following source data and figure supplement(s) for figure 3:

**Source data 1.** Immunofluorescence conditions in the experiment shown in *Figure 3A, E, F and G*.

**Source data 2.** Raw quantification data, immunofluorescence conditions, and detailed statistics of the experiment shown in *Figure 3B*.

**Source data 3.** Raw quantification data, immunofluorescence conditions, and detailed statistics of the experiment shown in *Figure 3C*.

**Source data 4.** Raw quantification data, immunofluorescence conditions, and detailed statistics of the experiment shown in *Figure 3D*.

**Figure supplement 1.** A potential problem of using MYO5A as a ciliary vesicle marker.

**Figure supplement 1—source data 1.** Immunofluorescence conditions of the experiment shown in *Figure 3—figure supplement 1*.

**Figure supplement 2.** RAB34 and MYO5A are recruited to the mother centriole earlier than EHD1 and PACSIN2.

**Figure supplement 2—source data 1.** Immunofluorescence conditions and raw quantification data of the experiment shown in *Figure 3—figure supplement 2A*.

**Figure supplement 2—source data 2.** Raw quantification data and detailed statistics of the experiment shown in *Figure 3—figure supplement 2E and F*.

**Figure supplement 2—source data 3.** Raw quantification data and detailed statistics of the experiment shown in *Figure 3—figure supplement 2G*.

**Figure supplement 2—source data 4.** Raw quantification data and detailed statistics of the experiment shown in *Figure 3—figure supplement 2H*.

**Figure supplement 3.** 3D structured illumination images of RAB34 and MYO5A.

**Figure supplement 3—source data 1.** Immunofluorescence conditions in the experiment shown in *Figure 3—figure supplement 3A–J*.

**Figure supplement 4.** Super-resolution reconstructions of RAB34 and MYO5A manually isolated from the data shown in *Figure 3H and J* with corresponding normalized histograms.

**Figure supplement 5.** Registration of the 3D single-molecule super-resolution data by imaging of FOP.

**Figure supplement 6.** Control of the registration of the 3D single-molecule super-resolution data by imaging of RAB34.

**Figure supplement 7.** Schematic of the optical setup used to collect 3D single-molecule super-resolution data.

knockout cells (*Figure 3—figure supplement 1C*). Note that the percentage of RAB34 and MYO5A double positive centrioles was highest in the confluent cells grown with serum and was decreased upon serum starvation (*Figure 3B*). We did not see the difference in the percentage of RAB34 positive centrioles between cells grown to 40–50% confluency (subconfluent) and 100% confluency (*Figure 3—figure supplement 2A*), confirming that the presence of RAB34 positive centrioles is not due to the confluency. The presence of the ciliary vesicle at the centriole before induction of cilium formation is consistent with the previous electron microscopy study (Figure 5D of *Insinna et al., 2019*). Vesicular fusion regulators, EHD1 and PACSIN2, were recruited to the centrosome at later time points after serum withdrawal (*Figure 3C*; *Figure 3D*; *Figure 3—figure supplement 2B–H*), confirming that both MYO5A and RAB34 are the earliest markers of the centriole-associated vesicle (likely the distal appendage vesicle) to date. Importantly, and in contrast to MYO5A, RAB34 did not localize to the pericentriolar region (*Figure 3—figure supplement 1A*). This makes RAB34 a more suitable vesicle marker to assess preciliary vesicle recruitment at the distal appendages. Consistent with this, *Rab34⁻/⁻* mice exhibit polydactyly and cleft palate as well as perinatal lethality, phenotypes reminiscent of cilia defects (*Xu et al., 2018*). This further supports a cilia-specific function of RAB34. In the 3D-SIM images, RAB34 localization was distal to CEP164 (*Figure 3E*; *Figure 3—figure supplement 3A*) and was displaced inwardly (*Figure 3F*; *Figure 3—figure supplement 3F*). This is reminiscent of the relative positioning between the vesicle and the distal appendages seen in electron microscopy (*Insinna et al., 2019*; *Lu et al., 2015*). The shape of RAB34 positive vesicles were highly variable even before the cilium formation was induced by serum starvation (*Figure 3—figure supplement 3G–J*). We also confirmed colocalization of MYO5A and RAB34 in 3D-SIM images (*Figure 3G*; *Figure 3—figure*

*supplement 3B–E*). To more precisely define the position of RAB34 in relation to MYO5A and to the mother centriole, we turned to two-color 3D single-molecule super-resolution imaging (*Bayas et al., 2019*; *Bennett et al., 2020*; *Gustavsson et al., 2018a*; *Gustavsson et al., 2018b*). This data showed that the MYO5A distribution was located at the edge of the RAB34 distribution on the vesicle (*Figure 3H–K*), with a 3D separation between the center of masses of the distributions of 89 nm and 67 nm in samples 1 and 2, respectively (*Figure 3—figure supplement 4*). We confirmed that the alternate localization of RAB34 and MYO5A is not due to channel registration by checking the complete colocalization of FOP between the two channels (*Figure 3—figure supplement 5*). We further confirmed this by testing colocalization of RAB34 stained with the two different secondary antibodies, Alexa Fluor 647 (AF647) and CF568 (*Figure 3—figure supplement 6*). The sizes of the RAB34 distributions, reflecting the measure of vesicle sizes, were found to be 230 nm x 170 nm x 190 nm for sample 1 and 190 nm × 170 nm × 250 nm for sample 2 reported as the $1/e^2$ of Gaussian fits of these distributions.

Collectively, these data suggest that RAB34 is a more specific marker for the centriole-associated vesicle (likely the distal appendage vesicle) than MYO5A and is located at a distinct position from MYO5A on the vesicle.

## Distal appendages independently regulate branched steps required for cilium formation

Distal appendages can minimally regulate four steps required for cilium formation: preciliary vesicle recruitment, IFT recruitment to the basal body, IFT initiation by recruiting CEP19-RABL2, and CP110 removal. We currently do not know whether these steps are independently regulated by distal appendages or are interconnected, so that the failure of one step may interrupt the subsequent steps of the cilium formation. In the latter case, only one of the four steps may be directly regulated by the distal appendages. To test this possibility, we inhibited one step at a time and tested if other steps are affected. To inhibit preciliary vesicle recruitment, we depleted RAB34 (*Figure 4—figure supplement 1A*), which was shown to inhibit formation/recruitment (*Xu et al., 2018*) or fusion (*Ganga et al., 2021*) of the preciliary/distal appendage vesicle. In contrast to MYO5A depletion, which did not affect ciliogenesis, depletion of RAB34 significantly inhibited the formation of the cilium (*Figure 4A*), as described (*Ganga et al., 2021*; *Oguchi et al., 2020*; *Stuck et al., 2021*; *Xu et al., 2018*). The absence of ciliation defects in *MYO5A* knockout cells differs from the previous observation (*Wu et al., 2018*), but is consistent with the fact that mutations in MYO5A gene cause Griscelli syndrome (*Pastural et al., 1997*), of which phenotypes are distinct from ciliopathies.

Electron microscopy analysis of RPE cells serum-starved for 3 hr revealed that only one out of 18 mother centrioles in RAB34-depleted cells had a vesicle, while 16 out of 41 control (sgGFP) cells had the vesicle(s) attached to centrioles (p<0.0054 in Fisher's exact test) (*Figure 4B–C*). This suggests that RAB34 is important for initial recruitment/formation of the vesicle, in agreement with the previous report (*Xu et al., 2018*). Whether RAB34 is also involved in the fusion of the vesicle at the later time point after serum starvation, as shown by the other report (*Ganga et al., 2021*), warrants further investigation. Using *RAB34* knockout cells, we tested whether disrupting the preciliary vesicle recruitment/formation affects the other steps of the cilium formation. Removal of CP110 from the mother centriole was modestly affected in *RAB34* knockout cells (*Figure 4D*), whereas IFT and CEP19 recruitment was not affected (*Figure 4E–F*), suggesting that preciliary vesicle recruitment is partially important to trigger CP110 removal. This result is inconsistent with two other studies, which showed no effect on CP110 removal in *RAB34* knockouts, potentially because of the difference in the duration of serum starvation (24 hrin our study versus 48 hr in the other studies *Ganga et al., 2021*; *Stuck et al., 2021*). We next tested whether recruitment of CEP19 or IFT affects preciliary vesicle recruitment. Note that CEP19 and IFT complex proteins localize to a compartment slightly below the distal appendage in the cells grown with serum, which infrequently show primary cilia (*Figure 1—figure supplement 5B*), while their localization is strongly enhanced upon serum starvation (*Figure 1—figure supplement 5C*; *Kanie et al., 2017*). To eliminate the IFT complexes or CEP19 from the mother centriole, we depleted either IFT52, a central component of the IFT-B complex (*Taschner et al., 2014*), or FGFR1OP (or FOP), which is required for centriolar localization of both IFT proteins and CEP19 (*Kanie et al., 2017*). As expected, IFT52 depletion greatly diminished the localization of all the other IFT complex proteins tested and thus inhibited the cilium formation (*Figure 4G–H*; *Figure 4—figure supplement 2A–E*).

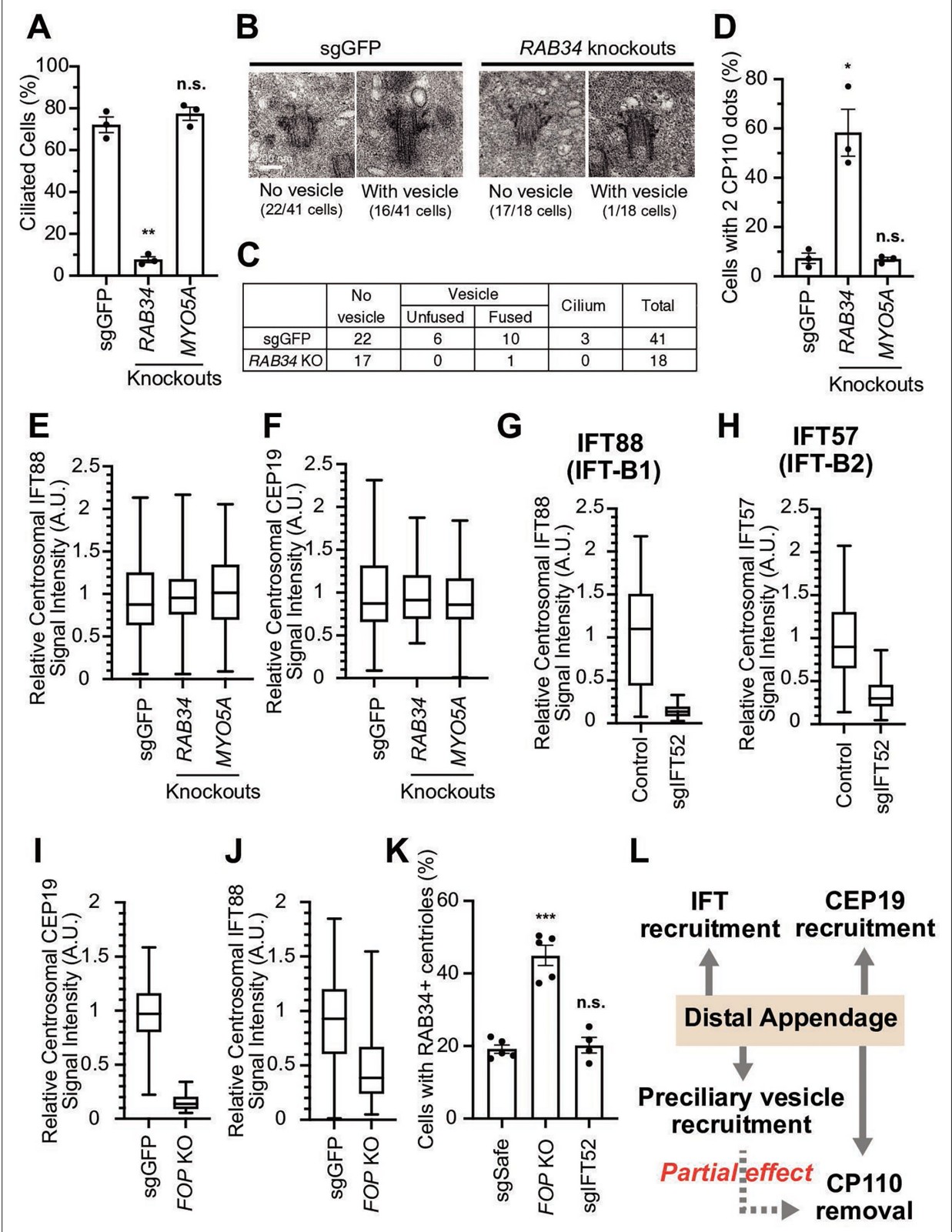

**Figure 4.** The distal appendage plays a role in preciliary vesicle recruitment, intraflagellar transport (IFT) recruitment, and CEP19 recruitment independently. (**A**) Cilium formation assay in control (sgGFP), *RAB34* knockout, or *MYO5A* knockout retinal pigment epithelial (RPE) cells serum starved for 24 hr. Data are averaged from three independent experiments, and each black dot indicates the value from an individual experiment. Error bars represent ± SEM. Statistics obtained through comparing between each knockout and control by Welch's t-test. The raw data, experimental conditions,

*Figure 4 continued on next page*

*Figure 4 continued*

and detailed statistics are available in *Figure 4—source data 1*. (**B**) Transmission electron microscopy analysis of the mother centriole in control (sgGFP) or *RAB34* knockout RPE cells serum starved for 3 hr. The representative images of the mother centrioles without (left) or with (right) a vesicle at the distal appendages are shown. (**C**) Quantification of the data shown in (**B**). The raw data and detailed statistics are available in *Figure 4—source data 2*. (**D**) CP110 removal assay in control (sgGFP), *RAB34* knockout, or *MYO5A* knockout RPE cells serum starved for 24 hr. Data are averaged from three independent experiments, and each black dot indicates the value from an individual experiment. Error bars represent ± SEM. Statistics obtained through comparing between each knockout and control by Welch's t-test. The raw data, experimental conditions, and detailed statistics are available in *Figure 4—source data 3*. (**E–J**) Box plots showing centrosomal signal intensity of IFT88 (E, G, and J), CEP19 (**F and I**), or IFT57 (**H**) in sgGFP control (E, F, I, and J), parental RPE-BFP-Cas9 control (**G and H**), indicated knockouts (E, F, I, and J), or RPE cells expressing sgIFT52 (**G and H**) serum starved for 24 hr. At least 40 cells were analyzed per each sample. The relative fluorescence signal intensity compared with the average of the control is shown. Data from a representative experiment are shown. The raw data and experimental conditions are available in *Figure 4—source data 4*, *Figure 4—source data 5*, *Figure 4—source data 6*, *Figure 4—source data 7*, *Figure 4—source data 8* and *Figure 4—source data 9*. (**K**) Preciliary vesicle recruitment assay in control (sgSafe) or indicated knockout RPE cells grown to confluent (without serum starvation). At least 90 cells were analyzed per each sample. The data is averaged from five independent experiments. Error bars represent ± SEM. Statistics obtained through comparing between each knockout and control by Welch's t-test. The raw data, experimental conditions, and detailed statistics are available in *Figure 4—source data 10*. (**L**) Summary of the role of the distal appendage. The distal appendage independently regulates IFT/CEP19 recruitment and preciliary vesicle recruitment, whereas CP110 removal is partially downstream of preciliary vesicle recruitment. A.U., arbitrary units; n.s., not significant; *p<0.05, **p<0.01, ***p<0.001.

The online version of this article includes the following source data and figure supplement(s) for figure 4:

**Source data 1.** Raw quantification data, immunofluorescence conditions and detailed statistics of the experiment shown in *Figure 4A*.

**Source data 2.** Raw quantification data and detailed statistics of the experiment shown in *Figure 4C*.

**Source data 3.** Raw quantification data, immunofluorescence conditions and detailed statistics of the experiment shown in *Figure 4D*.

**Source data 4.** Raw quantification data and immunofluorescence conditions of the experiment shown in *Figure 4E*.

**Source data 5.** Raw quantification data and immunofluorescence conditions of the experiment shown in *Figure 4F*.

**Source data 6.** Raw quantification data and immunofluorescence conditions of the experiment shown in *Figure 4G*.

**Source data 7.** Raw quantification data and immunofluorescence conditions of the experiment shown in *Figure 4H*.

**Source data 8.** Raw quantification data and immunofluorescence conditions of the experiment shown in *Figure 4I*.

**Source data 9.** Raw quantification data and immunofluorescence conditions of the experiment shown in *Figure 4J*.

**Source data 10.** Raw quantification data, immunofluorescence conditions and detailed statistics of the experiment shown in *Figure 4K*.

**Figure supplement 1.** Confirmation of *MYO5A* and *RAB34* knockouts by immunoblot.

**Figure supplement 1—source data 1.** The original files of the full raw unedited blots shown in *Figure 4—figure supplement 1A*.

**Figure supplement 1—source data 2.** The uncropped blots with boxes that indicate the regions displayed in *Figure 4—figure supplement 1A*.

**Figure supplement 2.** Characterization of IFT52 depleted cells.

**Figure supplement 2—source data 1.** The original files of the full raw unedited blots shown in *Figure 4—figure supplement 2A*.

**Figure supplement 2—source data 2.** The uncropped blots with boxes that indicate the regions displayed in *Figure 4—figure supplement 2A*.

**Figure supplement 2—source data 3.** Raw quantification data, immunofluorescence conditions, and detailed statistics of the experiment shown in *Figure 4—figure supplement 2B*.

**Figure supplement 2—source data 4.** Raw quantification data and immunofluorescence conditions of the experiment shown in *Figure 4—figure supplement 2C*.

**Figure supplement 2—source data 5.** Raw quantification data and immunofluorescence conditions of the experiment shown in *Figure 4—figure supplement 2D*.

**Figure supplement 2—source data 6.** Raw quantification data and immunofluorescence conditions of the experiment shown in *Figure 4—figure supplement 2E*.

Similarly, FOP depletion abrogated the localization of CEP19 and IFT88 as well as cilium formation (*Figure 4I,J*, *Figure 4—figure supplement 2B*). Depletion of neither FOP nor IFT52 disrupted the preciliary vesicle recruitment (*Figure 4K*), suggesting that preciliary vesicle recruitment can proceed independently of the IFT-CEP19 pathway. We currently do not know why the number of the vesicle-positive centrioles was increased in *FOP* knockout cells (*Figure 4K*). In summary, our data suggest that distal appendages independently regulate preciliary vesicle and IFT; CEP19 recruitment, whereas CP110 removal is partially downstream of preciliary vesicle recruitment (*Figure 4L*).

## CEP89 functions specifically in preciliary vesicle recruitment

We next sought to determine the function of each distal appendage protein. We first tested which distal appendage proteins play a role in cilium formation. The depletion of each component that is important for structural integrity of distal appendages (CEP83, SCLT1, CEP164, TTBK2) severely disrupted the cilium formation in either 24- or 48 hr serum-starved cells (*Figure 5A, B*). FBF1, CEP89, and ANKRD26 modestly affected cilium formation at 24 hr after serum removal (*Figure 5A*), but the ciliation defect in the knockout cells were ameliorated by prolonged incubation (48 hr) following serum starvation (*Figure 5B*). This suggests that these proteins are important for cilium formation, but that cells can compensate for the lack of these proteins and slowly catch up to form primary cilia. The distal appendage protein KIZ and LRRC45, as well as the distal appendage-associated protein, INPP5E, had no effect on cilium formation (*Figure 5A and B*). Ciliary length was mildly affected in *FBF1* and *INPP5E* knockout cells at the earlier time point (*Figure 5—figure supplement 1A*). Shorter ciliary length was observed in *ANKRD26* knockout cells serum starved for either 24 or 48 hr (*Figure 5—figure supplement 1A and B*). Interestingly, ARL13B signal intensity inside the cilium was diminished in *FBF1*, *CEP89*, or *ANKRD26* knockout cells even after the cells largely caught up on cilium formation after 48 hr of serum starvation (*Figure 5—figure supplement 1C and D*). This suggests that these knockouts can slowly form cilia, but the slowly formed cilia may not be functionally normal. The stronger defect in ciliary ARL13B signal in ANKRD26 might suggest a direct role of this protein in ARL13B recruitment around the distal appendages.

We next tested the importance of each distal appendage protein in preciliary vesicle recruitment. Consistent with their critical role in the structural integrity of the distal appendages, CEP83, SCLT1, CEP164, and TTBK2 severely disrupted preciliary vesicle recruitment (*Figure 5C*). Interestingly, CEP89 but not the other distal appendage proteins modestly but significantly affected preciliary vesicle recruitment (*Figure 5C*). The importance of CEP89 in preciliary vesicle recruitment is largely consistent with a previous report (*Sillibourne et al., 2013*). CP110 removal was again severely affected in knockouts of the four integral components of the distal appendages (CEP83, SCLT1, CEP164, and TTBK2) (*Figure 5D*). CEP89 depletion partially inhibited CP110 removal (*Figure 5D*), correlating with the partial effect on preciliary vesicle recruitment, which is upstream of CP110 removal (*Figure 4L*). IFT88 recruitment was severely disturbed in the knockouts of the four integral components (CEP83, SCLT1, CEP164, and TTBK2), but not in the other knockouts (*Figure 5E*). The effect of CEP164 or TTBK2 was slightly stronger than CEP83 or SCLT1, suggesting that CEP164 and TTBK2 may be more directly involved in this process.

CEP19 recruitment was strongly dependent on CEP164-TTBK2 (*Figure 5F*), but was only mildly affected by CEP83-SCLT1, which recruit CEP164 to the distal appendages (*Figure 2A and N*). This indicates that a very small amount of centriolar CEP164-TTBK2 may be sufficient to bring CEP19 near the distal appendage, and that CEP164-TTBK2, rather than CEP83-SCLT1, more directly regulates CEP19 recruitment.

In summary, the CEP83-SCLT1 structural module brings CEP164-TTBK2 to stabilize the distal appendages, and the CEP164-TTBK2 complex plays a more direct role in the cilium formation by regulating downstream processes including preciliary vesicle recruitment, IFT; CEP19 recruitment, and CP110 removal. In contrast, CEP89 is dispensable for structural integrity of the distal appendages, but it instead plays a crucial role in the preciliary vesicle recruitment (*Figure 6*).

## Discussion

Distal appendages are structures critical for the formation of the cilium. While their anatomical structure was described in the 1970s (*Anderson, 1972*; *Anderson and Brenner, 1971*), the first component, CEP164, was only found in 2007 (*Graser et al., 2007*). Since then, the list of distal appendage proteins has grown and the detailed protein architecture has been visualized by super-resolution microscopy (*Bowler et al., 2019*; *Yang et al., 2018*). Nevertheless, our understanding of the function of each distal appendage protein is limited.

In this study, we sought to comprehensively characterize previously known and newly identified distal appendage proteins (KIZ, NCS1, and CEP15) to deepen our understanding of the distal appendages. Among the three proteins that we identified, the last two factors will be described in detail in

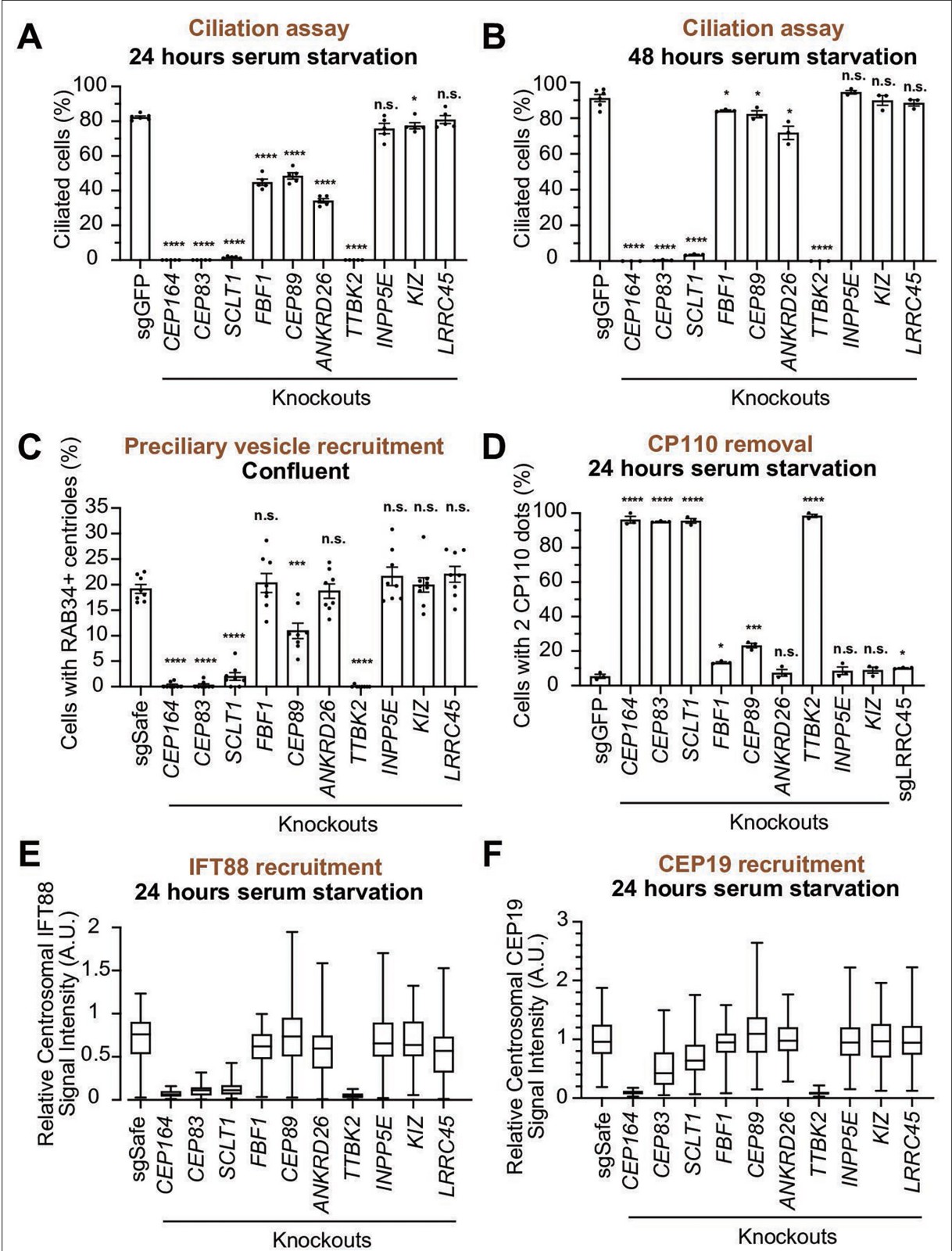

**Figure 5.** Functional analysis of distal appendage proteins reveals CEP89 as a protein important for preciliary vesicle recruitment. (**A–B**) Cilium formation assay in control (sgGFP) and indicated knockout retinal pigment epithelial (RPE) cells serum starved for 24 hr (**A**) or 48 hr (**B**). Data are averaged from five (**A**) or three (**B**) independent experiments, and each black dot indicates the value from an individual experiment. Error bars represent ± SEM. Statistics obtained through comparing between each knockout and control by Welch's t-test. The raw data, experimental conditions, and detailed statistics are

*Figure 5 continued on next page*

*Figure 5 continued*

available in *Figure 5—source data 1* and *Figure 5—source data 2*. (**C**) Preciliary vesicle recruitment assay in control (sgSafe) or indicated knockout RPE cells grown to confluent (without serum starvation). The data are averaged from eight independent experiments. Error bars represent ± SEM. Statistics obtained through comparing between each knockout and control by Welch's t-test. The raw data, experimental conditions, and detailed statistics are available in *Figure 5—source data 3*. (**D**) CP110 removal assay in control (sgGFP) and indicated knockout RPE cells serum starved for 24 hr. Data are averaged from three independent experiments, and each black dot indicates the value from an individual experiment. Error bars represent ± SEM. Statistics obtained through comparing between each knockout and control by Welch's t-test. The raw data, experimental conditions, and detailed statistics are available in *Figure 5—source data 4*. (**E–F**) Box plots showing centrosomal signal intensity of IFT88 (**E**) or CEP19 (**F**) in control (sgSafe) and indicated knockout RPE cells serum starved for 24 hr. The relative fluorescence signal intensity compared with the average of the control is shown. The data from a representative experiment are shown. The raw data and experimental conditions are available in *Figure 5—source data 5* and *Figure 5—source data 6*.A.U., arbitrary units; n.s., not significant; *p<0.05, **p<0.01, ***p<0.001.

The online version of this article includes the following source data and figure supplement(s) for figure 5:

**Source data 1.** Raw quantification data, immunofluorescence conditions, and detailed statistics of the experiment shown in *Figure 5A*.

**Source data 2.** Raw quantification data, immunofluorescence conditions, and detailed statistics of the experiment shown in *Figure 5B*.

**Source data 3.** Raw quantification data, immunofluorescence conditions, and detailed statistics of the experiment shown in *Figure 5C*.

**Source data 4.** Raw quantification data, immunofluorescence conditions, and detailed statistics of the experiment shown in *Figure 5D*.

**Source data 5.** Raw quantification data and immunofluorescence conditions of the experiment shown in *Figure 5E*.

**Source data 6.** Raw quantification data and immunofluorescence conditions of the experiment shown in *Figure 5F*.

**Figure supplement 1.** ARL13B intensity was reduced in the various distal appendage knockouts.

**Figure supplement 1—source data 1.** Raw quantification data, immunofluorescence conditions, and detailed statistics of the experiment shown in *Figure 5—figure supplement 1A*.

**Figure supplement 1—source data 2.** Raw quantification data, immunofluorescence conditions, and detailed statistics of the experiment shown in *Figure 5—figure supplement 1B*.

**Figure supplement 1—source data 3.** Raw quantification data, immunofluorescence conditions, and detailed statistics of the experiment shown in *Figure 5—figure supplement 1C*.

**Figure supplement 1—source data 4.** Raw quantification data, immunofluorescence conditions, and detailed statistics of the experiment shown in *Figure 5—figure supplement 1D*.

---

*Kanie et al., 2025* an accompanying paper (*Kanie et al., 2025*). A diagram summarizing the findings of our comprehensive analyses is shown in *Figure 6*.

## The structure of the distal appendages

The distal appendages consist of twelve proteins identified so far. Each protein localizes to a different positions at the distal appendages (*Figure 1B*). Interestingly, our current study revealed that CEP83, previously shown to locate at the innermost region of the distal appendages (*Bowler et al., 2019*; *Yang et al., 2018*), localize to the outermost region of the distal appendages when detected by antibodies that recognize a different epitope of CEP83 (*Figure 1A*). The further analyses using N-terminally and C-terminally GFP tagged CEP83 revealed that they localize to different position at the distal appendage (*Figure 1—figure supplement 4*). These results suggest that CEP83 has an extended structure that stretches the entire length of each blade of the distal appendages. This model is quite intriguing given that CEP83 is important for the localization of all the other distal appendage proteins (*Figure 2*) and its predicted structure, an extended alpha helix, would provide a perfect template to arrange components along the long axis of the distal appendage. The distal appendages may consist of stack of macromolecular complexes, each of which span the entire length of the structure. The model is also consistent with the previous observation, where C-terminal region of CEP164 locates closely at the wall of centriole and N-terminal part of the protein is spread to cover a wide region of the distal appendage (*Bowler et al., 2019*). Nonetheless, confirmation of this model needs additional investigation by crystallography and cryo-electron microscopy/tomography. SCLT1 localizes to the upper middle part of the distal appendages (*Figure 1A*) and was required for the localization of all the other distal appendage proteins (*Figure 2M and N*). SCLT1 also stabilizes CEP83, particularly at the outer region of the protein (*Figure 2C*; *Figure 6*). This suggests that CEP83-SCLT1 module works as a structural backbone of the distal appendages. Downstream of CEP83-SCLT1, we showed that CEP164-TTBK2 complex plays an important role in maintaining the distal appendage structure. Loss of TTBK2 resulted in the decrease in the localization of several distal appendage proteins, including

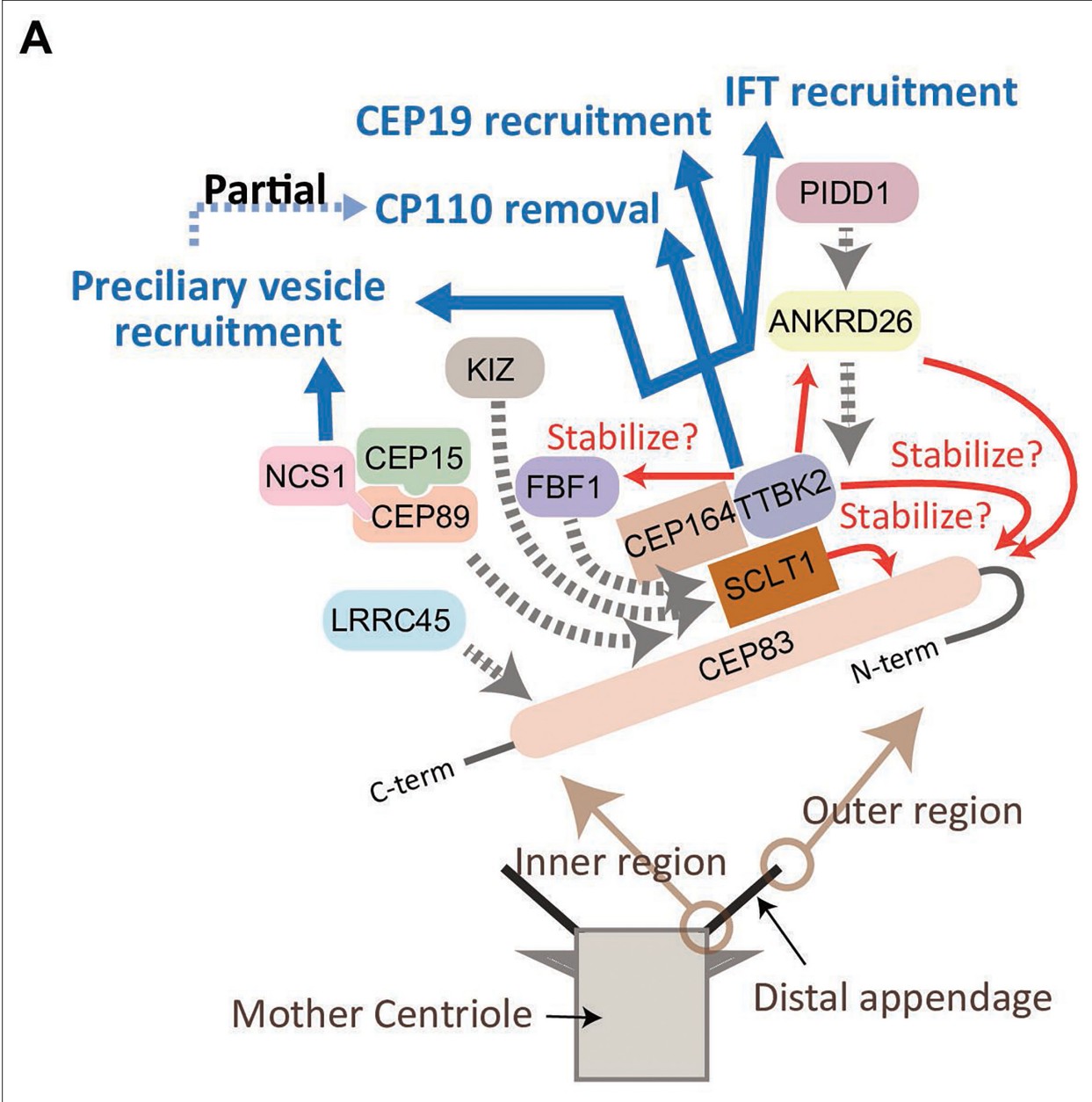

**Figure 6.** Model of the function of the distal appendage proteins. (**A**) CEP83 appears to form an extended structure that spans from the innermost region to the outermost region of the distal appendage to serve as a scaffold for the other distal appendages. SCLT1 stabilizes CEP83 especially at the outer region of the protein, and the CEP83-SCLT1 module recruits all the other distal appendage proteins. TTBK2 together with its upstream protein, CEP164, confers structural integrity to the distal appendage by stabilizing outer region of CEP83, and the other downstream proteins (ANKRD26, FBF1, and NCS1). The efficient localization of CEP164 to the distal appendage also requires TTBK2. ANKRD26 plays an important role in maintaining stability of outer region of CEP83, and recruits PIDD1. Kizuna (KIZ) is recruited to the distal appendage likely via direct interaction with SCLT1, as SCLT1 strongly affects protein stability of KIZ (*Figure 2—figure supplement 2*). LRRC45 is recruited to the innermost region of the distal appendage. The distal appendage is indispensable for cilium biogenesis by independently regulating IFT recruitment, CEP19 recruitment, CP110 removal, and preciliary vesicle recruitment, while the CP110 removal is partially downstream of preciliary vesicle recruitment. The CEP164-TTBK2 module may be the most critical regulator of these processes. The CEP89-CEP15-NCS1 module is recruited to the inner region of the distal appendage and is primarily important for the recruitment of the preciliary vesicle (described in greater detail in *Kanie et al., 2025* the accompanying paper). The red arrows indicate positive feedback; The blue arrows indicate functions achieved by the distal appendage; The dotted gray arrows indicate the recruitment of the proteins to the distal appendage.

the most upstream CEP83 (*Figure 2M and N*). This is consistent with the previous finding that TTBK2 phosphorylates distal appendage proteins, such as CEP83 and CEP89 (*Bernatik et al., 2020*; *Lo et al., 2019*). Further investigations are needed to assess whether other distal appendage proteins are also phosphorylated by TTBK2 and how the phosphorylation affects the structure and function of the substrates. ANKRD26 also affected the localization of several distal appendage proteins, including the outer ring of CEP83 (*Figure 2A, M and N*). These results emphasize that a set of proteins work together to organize the structure of distal appendages. Using the CRISPR-Cas9 system, we created a new hierarchical model of the distal appendage proteins, which is mostly consistent with the previously established model but has a number of modifications. Most of the discrepancies likely come from the differences between siRNA and CRISPR techniques. While CRISPR knockout lines may show genetic compensation, the overall consistency of our results with the previous CRISPR and siRNA results, in some cases showing only quantitative differences, do not suggest that genetic compensation is a major problem.

## The function of the distal appendages

Distal appendages are indispensable for cilium formation through regulation of at least four steps required for ciliogenesis: preciliary vesicle recruitment (*Schmidt et al., 2012*), recruitment of IFT (*Schmidt et al., 2012*) and CEP19-RABL2 (*Dateyama et al., 2019*), and CP110 removal (*Goetz et al., 2012*; *Tanos et al., 2013*). Our functional analyses revealed that all the four steps were almost completely disrupted when each of the four critical proteins (CEP83-SCLT1-CEP164-TTBK2) was depleted. This result might simply come from disorganization of the distal appendages, or those four proteins may be directly involved in the four ciliogenic processes. Since the IFT/CEP19 recruitment defect was milder in *CEP83/SCLT1* knockouts than *CEP164/TTBK2* knockouts, we predict that CEP164-TTBK2 complex may play more direct roles in the IFT/CEP19 recruitment. Cajánek and Nigg created a chimeric protein that consists of the distal appendage targeting region of CEP164 and the kinase domain of TTBK2. Intriguingly, this chimera was sufficient to almost fully rescue the ciliation defect of CEP164-depleted cells (Figure 5 of *Čajánek and Nigg, 2014*), suggesting that the main function of CEP164 is recruitment of TTBK2 to the distal appendages. Since the kinase activity of TTBK2 is critical for cilium formation (*Čajánek and Nigg, 2014*; *Goetz et al., 2012*), testing whether IFT complex proteins, CEP19, or their association partners (e.g. FGFR1OP) are phosphorylation targets of TTBK2 warrant future studies.

Our study also revealed CEP89 as a protein important for preciliary vesicle recruitment, but not for other processes of cilium formation. Given that CEP89 consists of two coiled-coil domains but lacks a membrane association domain, we hypothesized that CEP89 is involved in the preciliary vesicle recruitment via its interacting partner. Indeed, our further analysis revealed that Neuronal Calcium Sensor-1 (NCS1) interacts with CEP89 and is recruited to the distal appendages by CEP89. NCS1 then captures preciliary vesicles via its myristoylation motif. This story will be described in an accompanying paper (*Kanie et al., 2025*).

In addition to their critical functions in cilium formation, the distal appendages seem to play important roles in other biological processes. Recent studies showed that the activation of PIDD1 requires recruitment of the protein to the distal appendages by ANKRD26. The activated PIDD1 forms a complex, called PIDDsome, with CASP2 and CRADD. The PIDDsome then cleaves MDM2 and stabilizes p53 to inhibit proliferation in response to deleterious centrosomal amplification (*Burigotto et al., 2021*; *Evans et al., 2021*). ANKRD26 apparently has a dual function: cilia-related (stabilization of the outer ring of CEP83 and controlling ciliogenesis possibly via ARL13B regulation) and cilia-independent function (PIDD1 activation). It is currently unclear why PIDD1 activation occurs at the distal appendage. Nevertheless, it is possible that other distal appendage proteins may be involved in this process. It is also possible that distal appendages may be involved in yet unknown biological function.

Another possible role of the distal appendages is the regulation of the ciliary membrane protein composition. While there is no direct analysis, several lines of evidence support this hypothesis. First, distal appendages are located at the position where the mother centrioles attach to the plasma membrane, making it a strong candidate that modulates the composition of the ciliary membrane. Second, our current study showed that several distal appendage proteins have only modest or no effect on cilium formation (*Figure 5A and B*), but some of them are connected to ciliopathies (e.g.

KIZ (*El Shamieh et al., 2014*) and ANKRD26 *Acs et al., 2015*). Finally, the previous study showed that the ciliary G-protein coupled receptor, GPR161, is detained for a short period of time in the membrane compartment likely between the transition zone and the membrane anchor point of the distal appendage (*Ye et al., 2018*) before going out of the cilium in response to Hedgehog activation. This implies that the distal appendages might serve as a second diffusion or trafficking barrier besides the well-established transition zone (*Garcia-Gonzalo and Reiter, 2017*). Future studies will test this hypothesis and define the molecular mechanisms by which the distal appendages control ciliary membrane composition.

## Materials and methods

### Plasmids

pMCB306, a lenti-viral vector containing loxP-mU6-sgRNAs-puro resistance-EGFP-loxP cassette, and P293 Cas9-BFP were gifts from Prof. Michael Bassik. Lenti-virus envelope and packaging vector, pCMV-VSV-G and pCMV-dR8.2 dvpr, respectively, were gifts from Prof. Bob Weinberg (Addgene plasmid #8454 and #8455).

Lentiviral vectors containing single guide RNAs (sgRNAs) were generated by ligating 200 nM oligonucleotides encoding sgRNAs into the pMCB306 vector digested with BstXI (R0113S, NEB) and BlpI (R0585S, NEB) restriction enzymes. Before ligation, 4 µM of forward and reverse oligonucleotides listed in 'Source Data-Primers used for genomic PCR' were annealed in 50 µl of annealing buffer (100 mM potassium acetate, 30 mM HEPES (pH7.4), and 3 mM magnesium acetate) at room temperature following denaturation in the same buffer at 95 °C for 5 min. The targeting sequence for sgRNAs are listed in *Table 1*.

Gateway cloning compatible pDEST15PS vector used for bacterial protein expression was generated by inserting PreScission cleavage site immediately after GST tag into pDEST15 vector. pDEST15PS-ANKRD26 (214–537 a.a.) was generated by LR recombination between pENTR221-human ANKRD26 fragment (214–537 a.a.) and pDEST15PS vector.

Gateway cloning compatible lentiviral vectors, pWPXLd/LAP-N/blast/long EF/DEST and pWPXLd/LAP-C/blast/long EF/DEST vector were created by inserting N-terminally LAP tag (EGFP-TEV cleavage site-S tag-PreScission cleavage site)/DEST/blasticidin resistance cassette or DEST/C-terminally LAP tag/blasticidin resistance cassette into a second generation lenti-viral vector, pWPXLd. pWPXLd vector was a gift from Prof. Didier Trono (Addgene plasmid #12258).

The Gateway entry vector for Homo sapiens CEP83 lacking a stop codon was obtained from DNASU (HsCD00820673). The Gateway entry vector for *Homo sapiens* CEP83 containing STOP codon was generated via BP cloning using HsCD00820673 as a template.

A lentiviral vector containing N-terminally GFP tagged CEP83 was created by LR recombination between the CEP83 entry vector containing STOP codon and the pWPXLd/LAP-N/blast/long EF/DEST vector. A lentiviral vector containing C-terminally GFP tagged CEP83 was created by LR recombination between the CEP83 entry vector (HsCD00820673) and the pWPXLd/LAP-C/blast/long EF/DEST.

### Cell line, cell culture, transfection, and lentiviral expression

hTERT RPE-1 cells and 293T cells were grown in DMEM/F-12 (12400024, Thermo Fisher Scientific) supplemented with 10% FBS (100–106, Gemini), 1×GlutaMax (35050–079, Thermo Fisher Scientific), 100 U/mL Penicillin-Streptomycin (15140163, Thermo Fisher Scientific) at 37 °C in 5% $CO_2$. To induce cilium formation, cells were incubated in DMEM/F-12 supplemented with 1×GlutaMax and 100 U/mL Penicillin-Streptomycin (serum-free media). Both cell lines were authenticated via short-tandem-repeat based test. The authentication was performed by MTCRO-COBRE Cell line authentication core of the University of Oklahoma Health Science Center. Mycoplasma negativity of the original cell lines (hTERT RPE-1 and 293T) grown in antibiotics-free media were confirmed by a PCR-based test (G238, Applied Biological Materials).

The cell lines expressing sgRNA were generated using lentivirus. Lentivirus carrying loxP-mU6-sgRNAs-puro resistance-EGFP-loxP cassette was produced by co-transfecting 293T cells with 150 ng of pCMV-VSV-G, 350 ng of pCMV-dR8.2 dvpr, and 500 ng of pMCB306 plasmids described above along with 3 µl of Fugene 6 (E2692, Promega) transfection reagent. Media was replaced 24 hr after

transfection to omit transfection reagent, and virus was harvested at 48 hr post-transfection. Virus was then filtered with a 0.45 µm PVDF filter (SLHV013SL, Millipore) and mixed with fourfold volume of fresh media containing 12.5 µg/ml polybrene (TR-1003-G, Millipore). Following infection for 66 hr, cells were selected with 10 µg/ml puromycin (P9620, SIGMA-Aldrich).

The cell lines stably expressing GFP tagged CEP83 were generate using lentivirus with the method described above except that lenti-viral transfer plasmids (pWPXLd/LAP-N/blast/long EF/CEP83 and pWPXLd/LAP-C/blast/long EF/CEP83) instead of pMCB306 plasmids were used for transfection. The stable cell lines were selected with 10 µg/ml blasticidin (30–100-RB, Corning).

## CRISPR knockout

RPE cells expressing BFP-Cas9 (RPE-BFP-Cas9) were generated by infecting hTERT RPE-1 cells with lentivirus carrying P293 Cas9-BFP, followed by sorting BFP-positive cells using FACSAria (BD). RPE-BFP-Cas9 cells were then infected with lentivirus-carrying sgRNAs in the pMCB306 vector to generate knockout cells. After selection with 10 µg/ml puromycin, cells were subjected to immunoblotting, immunofluorescence, or genomic PCR combined with TIDE analysis (*Brinkman et al., 2014*) to determine knockout efficiency. The exact assay used for each cell line is listed in the CRISPR knockout cells summary (*Table 1A*). Cells were then infected with adenovirus carrying Cre-recombinase (1045 N, Vector BioLabs) at a multiplicity of infection of 50 to remove the sgRNA-puromycin resistance-EGFP cassette. 10 d after adenovirus infection, GFP-negative single-cells were sorted using FACSAria. The single-cell clones were expanded, and their knockout efficiency were determined by immunofluorescence, immunoblot, and/or genomic PCR (the detail described in the '*Table 1*'). The same number of validated single clones (typically three to four different clones) were mixed to create pooled single-cell knockout clones to minimize the phenotypic variability occurred in single-cell clones. The pooled clones were used in most of the experiments presented in this paper. The only exception is sgLRRC45 line used in *Figures 2H, I and 5D*, which are RPE-BFP-Cas9 cells infected with sgRNA followed by removal of sgRNA-puromycin resistance-EGFP cassette and GFP-negative bulk sorting (no single-cell cloning).

The targeting sequences of guide RNAs are listed in the *Table 1*.

## Transmission electron microscopy

Either control (sgGFP) or *RAB34* knockout RPE cells were grown on 12 mm round coverslips (12-545-81, Fisher Scientific), followed by serum starvation for 3 hr. Cells were then fixed with 4% PFA (433689 M, Alfa Aesar)/2% glutaraldehyde (G7526, SIGMA) in sodium cacodylate buffer (100 mM sodium cacodylate and 2 mM CaCl$_2$, pH 7.4) for 1 hr at room temperature, followed by two washes with sodium cacodylate buffer. Cells were then post-fixed in cold/aqueous 1% osmium tetroxide (19100, Electron Microscopy Sciences) in Milli-Q water for 1 hr at 4 °C, allowed to warm to room temperature (RT) for 2 hr rotating in a hood, and washed three times with Milli-Q water. The samples were then stained with 1% uranyl acetate in Milli-Q water at room temperature overnight. Next, the samples were dehydrated in graded ethanol (50%, 70%, 95%, and 100%), followed by infiltration in EMbed 812. Ultrathin serial sections (80 nm) were created using an UC7 (Leica, Wetzlar, Germany), and were picked up on formvar/Carbon coated 100 mesh Cu grids, stained for 40 s in 3.5% uranyl acetate in 50% acetone followed by staining in Sato's Lead Citrate for 2 min. Electron micrographs were taken on JEOL JEM1400 (120 kV) equipped with an Orius 832 digital camera with 9 µm pixels (Gatan). To test the percentage of the vesicle-positive centriole, multiple serial sections (typically 3–4) were analyzed per each mother centriole, as the vesicles are often not attached to all nine blades of the distal appendage (i.e. the vesicles are often not found in all the sections of the same mother centriole).

## Antibody generation

To raise rabbit polyclonal antibodies against ANKRD26, untagged human ANKRD26 fragments (214–537 a.a.) were injected into rabbits (1 mg for first injection and 500 µg for boosts). The ANKRD26 fragments were expressed as a GST fusion protein in Rosetta2 competent cells (#71402, Millipore) and purified using Glutathione Sepharose 4B Media (17075605, Cytiva) followed by cleavage of GST tag using GST tagged PreScission Protease (1 µg PreScission per 100 µg of recombinant protein). The

ANKRD26 antibody was affinity purified from the serum with the same antigen used for injection via standard protocols.

## Immunofluorescence

For wide-field microscopy, cells were grown on acid-washed 12 mm #1.5 round coverslips (72230–10, Electron Microscopy Sciences) and fixed either in 4% paraformaldehyde (433689 M, Alfa Aesar) in phosphate-buffered saline (PBS) for 15 min at room temperature or in 100% methanol (A412-4, Fisher Scientific) for 5 min at –20 °C. The primary antibodies used for immunofluorescence are listed in the 'Source Data-List of the antibodies -Distal appendage network-.' All staining conditions such as dilution of the antibodies can be found in the source data of each figure. After blocking with 5% normal serum that are matched with the species used to raise secondary antibodies (005-000-121 or 017-000-121, Jackson ImmunoResearch) in immunofluorescence (IF) buffer (3% bovine serum albumin (BP9703100, Fisher Scientific), 0.02% sodium azide (BDH7465-2, VWR International), and 0.1% NP-40 in PBS) for 30 min at room temperature, cells were incubated with primary antibody in IF buffer for at least 3 hr at room temperature, followed by rinsing with IF buffer five times. The samples were then incubated with fluorescent dye-labeled secondary antibodies (listed below) in IF buffer for 1 hr at room temperature, followed by rinsing with IF buffer five times. After nuclear staining with 4',6-diamidino-2-phenylindole (DAPI) (40043, Biotium) in IF buffer at a final concentration of 0.5 μg/ml, coverslips were mounted with Fluoromount-G (0100–01, SouthernBiotech) onto glass slides (3050002, Epredia). Images were acquired on an Everest deconvolution workstation (Intelligent Imaging Innovations) equipped with a Zeiss Axio Imager Z1 microscope and a CoolSnap HQ-cooled CCD camera (Roper Scientific). A 40 x NA1.3 plan-apochromat objective lens (420762–9800, Zeiss) was used for ciliation assays, and a 63 x NA1.4 plan-apochromat objective lens (420780–9900, Zeiss) was used for other analyses. All the raw image data are available through BioImage Archive (accession: S-BIAD1215; DOI: 10.6019/S-BIAD1215).

For ciliation assays, cells were plated into a six-well plate at a density of $2×10^5$ cells/well and grown for 66 hr. Cells were serum starved for 24 hr unless otherwise indicated and fixed in 4% PFA. After the blocking step, cells were stained for 3–4 hr at room temperature with anti-ARL13B (17711–1-AP, Proteintech), anti-CEP170 (41–3200, Invitrogen), and anti-acetylated tubulin (Ac-Tub) antibodies (T7451, SIGMA), washed, and then stained with anti-rabbit Alexa Fluor 488 (711-545-152, Jackson ImmunoResearch), goat anti-mouse IgG1-Alexa Fluor 568 (A-21124, Invitrogen), and goat anti-mouse IgG2b Alexa Fluor 647 (A-21242, Invitrogen). All the images were captured by focusing CEP170 without looking at a channel of the ciliary proteins to avoid selecting specific area based on the percentage of ciliated cells. The structures extending from the centrosome and positive for ARL13B with the length of more than 1 μm was counted as primary cilia. At least six images from different fields per sample were captured for typical analysis. Typically, at least 200 cells were analyzed per experiment. Exact number of cells that we analyzed in each sample can be found in the Source Data of corresponding figures. The percentage of ciliated cells were manually counted using the SlideBook software (Intelligent Imaging Innovations).

For preciliary vesicle recruitment assays, cells were plated into a 6-well plate at a density of $2×10^5$ cells/well, grown for 66 hr (without serum starvation), and fixed in 4% PFA. After the blocking step, cells were stained with anti-RAB34 (27435–1-AP, Proteintech), anti-Myosin Va (sc-365986, Santa Cruz), and anti-CEP170 (to mark centriole) antibodies (41–3200, Invitrogen), washed, then stained with goat anti-mouse IgG2a Alexa Fluor 488 (A-21131, Proteintech), goat anti-rabbit Alexa Fluor 568 (A10042, Invitrogen), and goat anti-mouse IgG1 Alexa Fluor 647 (A-21240, Invitrogen). All the images were captured by focusing CEP170 without looking at a channel of the vesicle markers to avoid selecting specific area based on the percentage of the vesicle-positive centrioles. At least eight images from different fields per sample were captured for typical analysis. Typically, at least 50 cells were analyzed per experiment. Exact number of cells that we analyzed in each sample can be found in the Source Data of corresponding figures.

For CP110 removal assays, cells were plated into a 6-well plate at a density of $2×10^5$ cells/well and grown for 66 hr. Cells were serum starved for 24 hr in 100% methanol. After the blocking step, cells were stained with anti-CP110 (12780–1-AP, Proteintech), anti-FOP (H00011116-M01, Abnova) (to mark both mother and daughter centrioles), and anti-CEP164 (sc-515403, Santa Cruz) (to mark the mother centriole) antibodies, washed, then stained with anti-rabbit Alexa Fluor 488 (711-545-152,

Jackson ImmunoResearch), goat anti-mouse IgG2a-Alexa Fluor 568 (A-21134, Invitrogen), and goat anti-mouse IgG2b Alexa Fluor 647 (A-21242, Invitrogen). All the images were captured by focusing FOP without looking at a channel of the other centriolar proteins to avoid selecting specific area based on the percentage of CP110 positive centrioles. CP110 localizing to both mother and daughter centrioles (as judged by colocalization with FOP) were counted as two dots, and CP110 localizing only to daughter centriole (as judged by no colocalization with CEP164) was counted as a one dot. Exact number of cells that we analyzed in each sample can be found in the Source Data of corresponding figures.

For structured illumination microscopy, cells were grown on 18 mm square coverslips with a thickness of 0.17 mm ±0.005 mm (474030-9000-000, Zeiss), fixed, and stained as described above. DAPI staining was not included for the structured illumination samples. Coverslips were mounted with SlowFade Gold Antifade Reagent (S36936, Life Technologies). Images were acquired either on a Nikon N-SIM-E/STORM super-resolution microscope (*Figure 1—figure supplement 4*) with a 100 x/1.49 NA CFI SR HP APO TIRF 100XC objective lens (Nikon) or on a DeltaVision OMX V4 system (*Figures 1A and 3F–H*, *Figure 1—figure supplement 1*, and *Figure 3—figure supplement 3*) equipped with a 100×/1.40 NA UPLANSAPO100XO objective lens (Olympus), and 488 nm (100 mW), 561 nm (100 mW), and 642 nm (300 mW) Coherent Sapphire solid state lasers and Evolve 512 EMCCD cameras (Photometrics). Image stacks of 2 µm z-steps were taken in either 0.1 µm (N-SIM-E/STORM system) or 0.125 µm increments (DeltaVision OMX) to ensure Nyquist sampling. Images were then computationally reconstructed and subjected to image registration by using NIS-Element (N-SIM-E/STORM system) or SoftWorx 6.5.1 software (DeltaVision OMX).

Secondary antibodies used for immunofluorescence were donkey anti-rabbit Alexa Fluor 488 (711-545-152, Jackson ImmunoResearch), donkey anti-mouse IgG DyLight488 (715-485-150, Jackson ImmunoResearch), goat anti-mouse IgG2a Alexa Fluor 488 (A-21131, Invitrogen), goat anti-mouse IgG1 Alexa Fluor 488 (A-21121, Invitrogen), donkey anti-Chicken IgY Alexa Fluor 488 (703-545-155, Jackson ImmunoResearch), donkey anti-rabbit IgG Alexa Fluor 568 (A10042, Invitrogen), goat anti-mouse IgG2a-Alexa Fluor 568 (A-21134, Invitrogen), goat anti-mouse IgG1-Alexa Fluor 568 (A-21124, Invitrogen), goat anti-mouse IgG2b Alexa Fluor 647 (A-21242, Invitrogen), goat anti-mouse IgG1 Alexa Fluor 647 (A-21240, Invitrogen), and donkey anti-rabbit IgG Alexa Fluor 647 (711-605-152, Jackson ImmunoResearch).

## Immunolabeling and sample preparation for 3D single-molecule super-resolution imaging

For 3D single-molecule super-resolution imaging, RPE-hTERT cells were plated in the central four wells of glass-bottom chambers (µ-Slide 8 Well, Ibidi) at the density of $3×10^4$ cells/well and grown for 48 hr in DMEM/F-12 supplemented with 10% FBS, 1×GlutaMax, and 100 U/mL Penicillin-Streptomycin at 37 °C in 5% $CO_2$. 24 hr before fixation, the medium was replaced with fresh DMEM/F-12 supplemented with 10% FBS, 1×GlutaMax and 100 U/mL Penicillin-Streptomycin. The cells were then fixed in 100% MeOH for 5 min at –20 °C. The slides were then washed twice in PBS and submerged and stored in PBS at 4 °C in Samco Bio-Tite sterile containers (010002, Thermo Scientific) until the day before imaging. Cells were permeabilized with three washing steps with 0.2% (v/v) Triton-X 100 in PBS with 5 min incubation between each wash and blocked using 3% bovine serum albumin (BSA, A2058, Sigma-Aldrich) in PBS for 1 hr at room temperature. In the experiments shown in *Figure 3H-K*, *Figure 3—figure supplement 4* and *Figure 3—figure supplement 5*, the cells were incubated with rabbit anti-RAB34 (27435–1-AP, Proteintech, 1:500), mouse IgG2a anti-MYO5A (sc-365986, Santa Cruz, 1:1000), and mouse IgG2b anti-FOP (H00011116-M01, Abnova, 1:1000) diluted in 1% BSA in PBS at 4 °C overnight, washed three times in 0.1% Triton-X 100 in PBS, and then incubated with donkey anti-rabbit Alexa Fluor 647 (ab150067, Abcam, 1:1000), goat anti-mouse IgG2b Alexa Fluor 647 (A-21242, Invitrogen, 1:1000), goat anti-mouse IgG2a CF568 (20258, Biotium, 1:1000), and goat anti-mouse IgG2b CF568 (20268, Biotium, 1:1000) diluted in 1% BSA in PBS for 1 hr shielded from light. In the experiments shown in *Figure 3—figure supplement 6*, the cells were incubated with rabbit anti-RAB34 (27435–1-AP, Proteintech, 1:500), mouse IgG2a anti-RAB34 (sc-365986, Santa Cruz, 1:250), and mouse IgG2b anti-FOP (H00011116-M01, Abnova, 1:1000) diluted in 1% BSA in PBS at 4 °C overnight, washed three times in 0.1% Triton-X 100 in PBS, and then incubated with donkey anti-rabbit Alexa Fluor 647 (ab150067, Abcam, 1:1000), goat anti-mouse IgG2b Alexa Fluor 647 (A-21242,

Invitrogen), goat anti-mouse IgG2a CF568 (20258, Biotium), and goat anti-mouse IgG2b CF568 (20268, Biotium) diluted in 1% BSA in PBS for 1 hr shielded from light. Then, the samples were washed five times with 0.1% Triton-X 100 in PBS, once with PBS, and stored in PBS at 4 °C while shielded from light until imaging up to several hours later. After aspiring remaining PBS, fluorescent beads (Tetra-Speck, T7280, 0.2 µm, Invitrogen, diluted 1:300 in Milli-Q water) were added to each well and allowed to settle for 10 min before being washed 10 x with PBS to remove unbound and loosely bound beads.

## Optical setup for 3D single-molecule super-resolution imaging

The optical setup was built around a conventional inverted microscope (IX83, Olympus) (*Figure 3—figure supplement 7*). Excitation lasers (560 nm and 642 nm, both 1000 mW, MPB Communications) were circularly polarized (LPVISC050-MP2 polarizers, Thorlabs; 560 nm: Z-10-A-.250-B-556 and 642 nm: Z-10-A-.250-B-647 quarter-wave plates, both Tower Optical) and filtered (560 nm: FF01-554/23-25 excitation filter, 642 nm: FF01-631/36-25 excitation filter, both Semrock), and expanded and collimated using lens telescopes. Collimated light was focused by a Köhler lens and introduced into the back port of the microscope through a Köhler lens to allow for wide-field epi-illumination. The lasers were toggled with shutters (VS14S2T1 with VMM-D3 three-channel driver, Vincent Associates Uniblitz).

The sample was positioned on an xy translation stage (M26821LOJ, Physik Instrumente) and an xyz piezoelectric stage (P-545.3C8H, Physik Instrumente). The emission from the sample was collected using a high numerical aperture (NA) objective (UPLXAPO100XO, 100 x, NA 1.45, Olympus) and filtered (ZT405/488/561/640rpcV3 dichroic; ZET561NF notch filter; and ZET642NF notch filter, all Chroma) before entering a 4 f imaging system. The first lens of the 4 f imaging system ($f$=80 mm, AC508-080-AB, Thorlabs) was placed one focal length from the intermediate image plane in the emission path. A dichroic mirror (T660lpxr-UF3, Chroma) was placed after the first 4 f lens in order to split the light into two different spectral paths, where far red light ('red channel') was transmitted into one optical path and greener light ('green channel') was reflected into the other optical path. In order to reshape the point spread function (PSF) of the microscope to encode the axial position (z) of the individual fluorophores, transmissive dielectric double helix (DH) phase masks with ~2 µm axial range (green channel: DH1-580-3249, red channel: DH1-680-3249, both Double Helix Optics) were placed one focal length after the first 4 f lens in each path and another 4 f lens was placed one focal length after the phase masks in both paths. Bandpass filters (red channel: two ET700/75 m bandpass filters; green channel: ET605/70 m bandpass filter, both Chroma) were placed in the paths between the phase masks and the second 4 f lenses. The second 4 f lenses then focused the light onto an EM-CCD camera (iXon Ultra 897, Andor) placed one focal length away from the second 4 f lenses.

## Two-color 3D single-molecule super-resolution imaging

To facilitate calibration of the engineered PSFs and registration between the two channels, a solution of fiducial beads (TetraSpeck, T7280, 0.2 µm, Invitrogen) were diluted 1:5 in 10% polyvinyl alcohol (Mowiol 4–88, 17951, Polysciences Inc) in Milli-Q water and spun-coat onto plasma-cleaned coverslips (#1.5 H, 22x22 mm, 170±5 µm, CG15CH, Thorlabs). For calibration of the PSFs, scans over a 2 µm axial range with 50 nm steps were acquired using the piezoelectric xyz translation stage. For registration measurements, the stage was translated in xy to ten different positions, and stacks of 50 frames were acquired at each position. Dark frames (400) were collected with the camera shutter closed before image acquisition, and the averaged intensity was subtracted from the calibration, registration, and single-molecule data before further analysis.

Directly prior to cell imaging, a reducing and oxygen-scavenging buffer optimized for dSTORM blinking (*Halpern et al., 2015*) comprising 100 mM Tris-HCl (pH 8, J22638-K2, Thermo Scientific), 10% (w/v) glucose (215530, BD Difco), 2 µl/ml catalase (C100, Sigma-Aldrich), 560 µg/ml glucose oxidase (G2133, Sigma-Aldrich), and 143 mM β-mercaptoethanol (M6250, Sigma-Aldrich) was added to the well and the well was sealed with parafilm. The samples were then kept in this buffer both for diffraction-limited imaging and single-molecule imaging.

For diffraction-limited imaging, cells were imaged using laser intensities of 0.3 W/cm$^2$ for the 642 nm laser and 1.2 W/cm$^2$ for the 560 nm laser. Before beginning the single-molecule super-resolution imaging, a large fraction of the fluorophores in the field of view were converted into a dark state using 560 nm and 642 nm illumination each at ~5 kW/cm$^2$. The same laser intensities were then

used for sequential acquisition of 100,000 frames of single-molecule data in each channel, first using exposure times of 50 ms for imaging of AF647 fluorophores in the red channel and then 35 ms for imaging of CF568 fluorophores in the green channel using calibrated EM gain and conversion gain of the camera of 183 and 4.41 photoelectrons/ADC count, respectively. Fiducial beads were detected in each frame to facilitate drift correction in post-processing.

## Analysis of 3D single-molecule super-resolution data

Images with acquired data from the two channels were cropped in ImageJ before analysis. Stacks that were acquired of fiducial beads for calibration were averaged over 50 frames at each unique position. These DH PSF calibration scans and the single-molecule images were used for calibration and localization using fit3Dspline in the modular analysis platform SMAP (*Ries, 2020*). Filter sizes and intensity count cutoffs, which serve as a threshold for template matching, were adjusted between samples based on localization previews to maximize correct localizations and minimize mislocalizations as identified by eye. Sample drift during image acquisition was accounted for by localizing fiducial beads in the same field of view as the single-molecule data. The measured motion of the fiducial bead was smoothed via cubic spline fitting (MATLAB function *csaps* with a smoothing parameter of $10^{-6}$) and subtracted from the single-molecule data using custom-written MATLAB scripts.

Registration between the two-color channels was completed in the x- and y-directions before correcting the z-direction. Images of dense fluorescent beads spun onto a coverslip were acquired at ten different xy positions and averaged in ImageJ. This averaged image was then cropped into the same fields of view as used for data acquisition in the two channels, and the MATLAB function *imregtform* was used to find the affine transformation that mapped the beads in the green channel onto the beads in the red channel. As the registration data was acquired at the coverslip while the single-molecule data could be acquired multiple microns above the coverslip through the cell, the registration was fine-tuned in an additional step by adapting a 2D cross-correlation approach (*Schnitzbauer et al., 2018*) to 3D and using it to account for any residual nanoscale offsets caused by aberrations when imaging higher up in the sample and to correct for any offset in the z-direction. The protein FOP is known to localize to the region close to the subdistal appendages of the mother centriole and daughter centriole, and forms ring-like structures (*Kanie et al., 2017*). By labeling these FOP structures with both AF647 and CF568, they served as a ground-truth for fine-tuning the channel registration. The FOP structures at the mother and daughter centrioles were manually isolated in each channel using Vutara SRX (version 7.0.00, Bruker) and cross-correlation was used to maximize the colocalization of the FOP localizations in the two registered channels (*Figure 3—figure supplement 6*). This cross-correlation was performed over multiple iterations until the shift between the two-color channels was below 2 nm for each axis. The translational shift applied to the red channel that yielded the maximum colocalization coefficient between the FOP structures in the two channels was then applied to all single-molecule localizations in the region of interest, thereby providing a nanoscale fine-tuning of the registration (*Figure 3—figure supplements 5 and 6*).

Following calibration, localization, drift correction, and registration of the data, localizations were rendered in Vutara SRX, where each localization was represented by a 3D Gaussian with 20 nm diameter and with variable opacities set to best visualize the localization density. The localizations were filtered to remove localizations with xy Cramér-Rao Lower Bound (CRLB) values from SMAP below 20 nm, and spurious localizations were removed by means of filtering for large average distance to eight nearest neighbors. This resulted in reconstructions containing the following number of localizations: sample 1, RAB34: 12276, MYO5A: 1867, and FOP: 16093; sample 2, RAB34: 7239, MYO5A: 588, and FOP: 14255. The opacities used for visualization are as follows: sample 1, RAB34: 0.03, MYO5A: 0.08, FOP: 0.05; sample 2, RAB34: 0.02, MYO5A: 0.09, FOP: 0.05 (*Figure 3H and J*).

Sizes of the RAB34 distributions were found by plotting histograms of the localizations along the x, y, and z axes and extracting the $1/e^2$ values from their Gaussian fits. Threshold values for the localizations to be counted as a vesicle candidate were set at 120 nm in each direction, and clusters of localizations that had smaller dimensions were excluded. This threshold was based on the dimensions of clusters of localizations many micrometers away from the centrioles that were not likely RAB34 on vesicles compared to the dimensions of vesicle candidates within a micron from the centrioles. For RAB34-MYO5A offset measurements, histograms of the localizations along the x, y, and z axes for both RAB34 and MYO5A were fitted to two Gaussians to estimate the center-of-mass (COM)

separation. The difference between the centers of the two peaks yielded the offset between RAB34 and MYO5A in each dimension, and the total distance was calculated in 3D. Distances of the RAB34 and MYO5A structures from the mother centriole FOP structure were defined by the difference in 3D distance between the COM of the FOP structure and the COM of RAB34 and MYO5A, respectively.

### Immunoblot

For immunoblotting, cells were grown to confluent in a 6-well plate and lysed in 100 µl of NP-40 lysis buffer 50 mM Tris-HCl [pH7.5], 150 mM NaCl, and 0.3% NP-40 (11332473001, Roche Applied Science) containing 10 µg/ml LPC (leupeptin, pepstatin A, and chymostatin), and 1% phosphatase inhibitor cocktail 2 (P5726, SIGMA) followed by clarification of the lysate by centrifugation at 15,000 rpm (21,000 g) at 4 °C for 10 min. 72.5 µl of the clarified lysates were then mixed with 25 µl of 4×Lithium Dodecyl Sulfate (LDS) buffer (424 mM Tris-HCl, 564 mM Tris-base, 8% LDS, 10% glycerol, 2.04 mM EDTA, 0.26% Brilliant Blue G250, 0.025% phenol red) and 2.5 µl of 2-mercaptoethanol (M3148, SIGMA), and incubated at 95 °C for 5 min. Proteins were separated in NuPAGE Novex 4–12% Bis-Tris protein gels (WG1402BOX, Thermo Fisher Scientific) in NuPAGE MOPS SDS running buffer (50 mM MOPS, 50 mM Tris Base, 0.1% SDS, 1 mM EDTA, pH 7.7), then transferred onto Immobilon-FL PVDF Transfer Membranes (IPFL00010, EMD Millipore) in Towbin Buffer (25 mM Tris, 192 mM glycine, pH 8.3). Membranes were incubated in LI-COR Odyssey Blocking Buffer (NC9232238, LI-COR) for 30 min at room temperature, and then probed overnight at 4 °C with the appropriate primary antibody diluted in blocking buffer. Next, membranes were washed 3×5 min in TBST buffer (20 mM Tris, 150 mM NaCl, 0.1% Tween 20, pH 7.5) at room temperature, incubated with the appropriate IRDye antibodies (LI-COR) diluted in blocking buffer for 30 min at room temperature, then washed 3×5 min in TBST buffer. Membranes were scanned on an Odyssey CLx Imaging System (LI-COR) and proteins were detected at wavelengths 680 and 800 nm.

Primary antibodies used for immunoblotting are listed in the 'Source Data-List of the antibodies -Distal appendage network-.' Secondary antibodies used for immunoblotting were IRDye 800CW donkey anti-rabbit (926–32213, LI-COR) and IRDye 680CW donkey anti-mouse (926–68072, LI-COR).

### Sequence alignment

Protein sequence alignment shown in *Figure 1—figure supplement 2A* was performed via global alignment with free end gaps using BLOSUM 62 matrix on the Geneious Prime Software.

### Experimental replicates

The term 'replicates' used in this paper indicates that the same cell lines were plated at different dates for each experiment. In most cases, cell lines were thawed from liquid nitrogen at different dates and immunostaining was performed at different dates among the replicates.

### Quantification of fluorescence intensity and statistical analysis

#### Fluorescence intensity measurements

The fluorescence intensity was measured with 16-bit TIFF multi-color stack images acquired at 63 x magnification (NA1.4) by using Image J software. All the images for measurement of centrosomal signal intensity were captured by focusing CEP170 without looking at a channel of the protein of interest (POI) to avoid selecting specific area based on the signal intensity of POI. To measure the fluorescence intensity of centrosomal proteins, channels containing CEP170 and the protein of interest (POI) were individually extracted into separate images. A rolling ball background subtraction with a rolling ball radius of 5 pixels was implemented for both CEP170 and the POI to perform local background subtraction. The mask for both CEP170 and the POI was created by setting the lower threshold to the minimum level that covers only the centrosome. Each mask was then combined by converting the two masks to a stack followed by z projection. The combo mask was then dilated until the two masks were merged. After eroding the dilated masks several times, the fluorescent intensity of the POI was measured via 'analyze particles' command with optimal size and circularity. The size and circularity are optimized for individual POI to detect most of the centrosome in the image without capturing non-centrosomal structure. Outliers (likely non-centrosomal structure) were then excluded from the data using the ROUT method with a false discovery rate of 1% using GraphPad Prism 9 software to further remove signals from non-centrosomal structures, as the signals from those are typically

extremely lower or higher than those from centrioles. Fluorescence intensity of ciliary proteins were measured similarly to centrosomal proteins but with several modifications. A mask was created for only ARL13B by setting the lower threshold to the minimum level that covers only cilia. The size and circularity are optimized for individual POI to detect only cilia without capturing non-ciliary structure. Image macros used for the automated measurement described above are found in the supplementary files.

To test whether the difference in the signal intensity is statistically different between control and test samples, the intensity measured through the described method was compared between control and test samples using nested one-way ANOVA with Dunnett's multiple comparisons test if there are more than two replicates. In case there are less than three replicates, the statistical test was not performed in a single experiment, as the signal intensity is affected slightly by staining procedure and statistical significance is affected largely by the number of cells examined. For example, we saw statistical significance in the signal intensity with the same samples that are stained independently if we analyze the large number of the cells (more than 100 cells). Instead, we confirmed the same tendency in the change of fluorescence intensity in the test samples across two replicates.

### Statistical analysis for ciliation, preciliary vesicle recruitment, and CP110 removal assay

For ciliation, preciliary vesicle recruitment, and CP110 removal assay, the number of ciliated cells from the indicated number of replicates were compared between control (sgGFP or sgSafe) and the test samples using Welch's t-test. The exact number of samples and replicates are indicated in the resource data of the corresponding figures.

### Statistical analysis for ciliary length measurements

For ciliary length measurements, shown in *Figure 5—figure supplement 1A and B*, the ciliary length was compared between control and the knockouts using nested t-test.

### Diameter measurements of the distal appendage rings

For diameter measurements of the ring shown in *Figure 1C*, *Figure 1—figure supplement 4*, maximal intensity projection with top view images of the centriolar ring were first carried out and the peak-to-peak diameter was measured from four different angles and averaged to reduce the variability caused by tilting of the centriole. The number of the top view of the distal appendage rings analyzed is indicated in *Figure 1—source data 1C–source data*.

For all the statistics used in this paper, asterisks denote $*0.01 \leq p < 0.05$, $**p < 0.01$, $***p < 0.001$, n.s.: not significant. All the statistical significance was calculated by using GraphPad Prism 9 software.

### Materials availability statement

All the newly created materials used in this paper including ANKRD26 antibody, plasmids, stable cell lines are readily available from the corresponding authors (Tomoharu-Kanie@ouhsc.edu or pjackson@stanford.edu) upon request.

## Acknowledgements

We thank Drs. Bahtiyar Kurtulmus and Gislene Pereira for LRRC45 antibodies, and Dr. Steve Caplan for the EHD1 antibody. We thank Dr. Jonathan Mulholland for technical advice on the 3D-SIM experiments. We thank Mr. John Perrino for his technical support in sample preparation for the electron microscopy experiments. We thank members of the Jackson lab for helpful discussion and advice, especially Dr. Markus Kelly for establishing the method for semi-automated measurement of centrosomal signal intensity. We thank Ms. Sofía Vargas-Hernández for her help in developing the 3D cross-correlation code for the single-molecule data. 3D-SIM experiments were performed at the Stanford Cell Sciences Imaging Facility (DeltaVision OMX V4) and at the BMSB Imaging Core of the University of Oklahoma Health Sciences Center (Nikon N-SIM-E/STORM). DeltaVision OMX V4 was supported by Award Number 1S10OD01227601 from the National Center for Research Resources (NCRR). The Nikon N-SIM-E/STORM super-resolution microscope is supported by a Large Equipment Grant from the Oklahoma Center for Adult Stem Cell Research (OCASCR) and the OUHSC Department of Cell

Biology. Electron microscopy observation was performed at the Stanford Cell Sciences Imaging Facility and were supported by NIH S10 Award Number 1S10OD028536-01, titled 'OneView 4kX4k sCMOS camera for transmission electron microscopy applications.' The cell authentication service performed by MTCRO-COBRE Cell line authentication core of the University of Oklahoma Health Science Center was supported partly by P20GM103639 and National Cancer Institute Grant P30CA225520 of the National Institutes of Health (NIH). This project was supported by funds from the Baxter Laboratory for Stem Cell Research, the Stanford Department of Research, the Stanford Cancer Center, NIH grants R01GM114276 and R01GM121565 to PKJ, NIH grant P20GM103447 and 1R35GM151013 to TK, and partial financial support from the National Institute of General Medical Sciences of the National Institutes of Health grant R00GM134187 and grant R35GM155365, the Welch Foundation grant C-2064–20210327, and startup funds from the Cancer Prevention and Research Institute of Texas grant RR200025 to AKG.

## Additional information

### Funding

| Funder | Grant reference number | Author |
|---|---|---|
| National Institute of General Medical Sciences | P20GM103447 | Tomoharu Kanie |
| National Institute of General Medical Sciences | 1R35GM151013 | Tomoharu Kanie |
| National Institute of General Medical Sciences | R00GM134187 | Anna-Karin Gustavsson |
| Welch Foundation | C-2064-20210327 | Anna-Karin Gustavsson |
| Cancer Prevention and Research Institute of Texas | RR200025 | Anna-Karin Gustavsson |
| National Institute of General Medical Sciences | R01GM114276 | Peter K Jackson |
| National Institute of General Medical Sciences | R01GM121565 | Peter K Jackson |
| National Institute of General Medical Sciences | R35GM155365 | Anna-Karin Gustavsson |

The funders had no role in study design, data collection and interpretation, or the decision to submit the work for publication.

### Author contributions

Tomoharu Kanie, Conceptualization, Resources, Data curation, Formal analysis, Supervision, Funding acquisition, Validation, Investigation, Visualization, Methodology, Writing – original draft, Project administration, Writing – review and editing; Beibei Liu, Formal analysis, Investigation, Methodology, Writing – review and editing; Julia F Love, Saxton D Fisher, Data curation, Formal analysis, Investigation, Visualization, Methodology, Writing – review and editing; Anna-Karin Gustavsson, Formal analysis, Supervision, Funding acquisition, Investigation, Methodology, Writing – review and editing; Peter K Jackson, Conceptualization, Resources, Supervision, Funding acquisition, Investigation, Methodology, Project administration, Writing – review and editing

### Author ORCIDs

Tomoharu Kanie ⑩ https://orcid.org/0000-0002-2084-1451
Julia F Love ⑩ https://orcid.org/0000-0001-6642-6301
Anna-Karin Gustavsson ⑩ https://orcid.org/0000-0002-0980-1168

### Decision letter and Author response

Decision letter https://doi.org/10.7554/eLife.85999.sa1
Author response https://doi.org/10.7554/eLife.85999.sa2

# Additional files

## Supplementary files
MDAR checklist

Source data 1. the full raw unedited immunoblot with label.

Source data 2. Macro for measuring fluorescent intensity of centrosomal proteins.

Source data 3. Macro for measuring fluorescent intensity of ciliary proteins.

Source data 4. List of cell lines used in this paper.

Source data 5. List of the antibodies -Distal appendage network-.

Source data 6. Primers used for genomic PCR.

## Data availability
All the raw image data shown or analyzed in this paper are available through BioImage Archive (accession: S-BIAD1215; DOI: 10.6019/S-BIAD1215). Macro for automated fluorescent intensity measurement are included in the supporting files; Source Data files have been provided for all the corresponding figures.

The following dataset was generated:

| Author(s) | Year | Dataset title | Dataset URL | Database and Identifier |
|---|---|---|---|---|
| Kanie T | 2024 | A hierarchical pathway for assembly of the distal appendages that organize primary cilia | https://doi.org/10.6019/S-BIAD1215 | BioImage Archive, 10.6019/S-BIAD1215 |

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
