## [Editor Report]

This fundamental study presents the most comprehensive view of the functional organization and requirements for a mother centriole's distal appendage in primary cilia assembly published to date. Crispr-knockouts and super-resolution microscopy analysis of the distal appendage proteins provides compelling evidence to support the claims of the authors. This work will be of high value to cell biologists and biophysicists working on the structure and function of the centrosome as well as human geneticists exploring ciliary pathology.

---

## [Decision Letter]

**Decision letter after peer review:**

Thank you for submitting your article "A hierarchical pathway for assembly of the distal appendages that organize primary cilia" for consideration by *eLife*. Your article has been reviewed by 3 peer reviewers, including Gregory J Pazour as the Reviewing Editor and Reviewer #1, and the evaluation has been overseen by Piali Sengupta as the Senior Editor.

We were all impressed with the extensive amount of analysis that was performed. The largest concern that was raised during the review and in the post-review discussion involved the lack of rescue of knockout cell lines to ensure that the phenotypes are due to the gene of interest. In most cases, I do not feel this is required, however, Cep83 is an exception. The intriguing conclusion that Cep83 is an extended molecule linking distant parts of the appendage is based on two antibodies with extensive cross-reactivity to other proteins. Further work with tagged rescue constructs is needed to support this conclusion. The additional concerns raised by the reviewers can be addressed by text revisions but we also suggest that you may want to consider adding data to address a subset of the concerns raised.

*Reviewer #1 (Recommendations for the authors):*

A major new finding of the work is that Cep83 is an extended protein that provides the backbone for much of the distal appendage. This conclusion is drawn from the localization of two antibodies that show extensive cross-reactivity to a ladder of cellular proteins. While I appreciate that the signal from these antibodies is reduced at the centrosome in the Cep83 knockout lines, alternative explanations could be made. To strengthen this conclusion, the authors should rescue the knockout cell line with constructs tagged at various positions. Representative images used for the quantification in Figure 2 should be shown.

The model presented in figure 6 appears to be more of a visual abstract rather than a diagram that would help to summarize the findings of the paper. Even after reading the paper and studying the figure legend, I have little understanding of what figure 6 is trying to show. This paper would have more impact if the figure showed the main points of the document. That being diagrams illustrating how the proteins are organized with respect to each other and another of the hierarchy of assembly.

*Reviewer #2 (Recommendations for the authors):*

There are a number of points in this paper.

1. Throughout the paper, there is a lack of recognition of the Tanos et al. paper, which initially described a number of points discussed in this paper. Although Tanos et al. is cited, the work not only identified four novel distal appendage proteins but also molecularly showed how distal appendages mediated centriole to membrane docking through the recruitment of TTBK2 and the removal of CP110 and the recruitment of IFT molecules. In addition, her careful EM analysis of polarized IMCD3 cells using serial sections transversally cut along an apicobasal axis, and TEM clearly showed that, in the absence of CEP83, centrioles did not dock to the plasma membrane. I assume that this is an unintentional omission and I have pointed out specific points below.

2. To map the localization of centriole distal appendage proteins, the authors use 3D-SIM. Several studies have used STORM, correlated EM, and expansion microscopy to look at the structure and localization of distal appendages. This includes Yang et al. (PMID: 29789620), Bowler et al. (PMID: 30824690), and others. It, therefore, does not seem conclusive to use an inferior technique that is not properly validated by correlated EM. Controls with previously used techniques have not been carried out. Could additional cell lines be used to test these antibodies? Antibodies for centriolar proteins have been shown to frequently result in unspecific centriolar signals. Could these new antibodies be tested for immunostaining in knockout cell lines to see if the centriolar signal is specific?

3. Most of the DAP localization data shown is not new. However, three novel distal appendage proteins are described. Regarding the CEP83 localization, a novel localization in the outer rings of DAPs is found. Additional techniques should be used to confirm this. Furthermore, a number of antibodies for centriolar proteins show unspecific signals at centrioles. Has this antibody staining been tested in CEP83 knockout cells? A western blot is shown for the knockout cells but not for immunostaining. In "Figure 1—figure supplement 2 Characterization of the two CEP83 antibodies", here, the western blot is showing a band that seems to run at more than 90 KDa, the band does not seem to correspond to CEP83 and seems very unspecific. Has this localization been validated using EM? 3D-SIM has also been shown to generate artifacts, but this has not been taken into account.

4. Regarding this, please see the points below, line by line:

Line 31 "Functional assay revealed that CEP89 selectively functions in RAB34+ ciliary vesicle

Recruitment", should say "functional assays". Please correct the language throughout.

Line 50 "Tsou, 2016?" what reference is this? Is this in the references section?

Line 54 says: "2 recruitment of intra-flagellar transport (IFT) protein complexes (Cajanek and Nigg, 2014; Goetz, Liem, and Anderson, 2012; Schmidt et al., 2012)"

Here, Tanos et al. should be cited (PMID: 23348840), as this was also shown in the Tanos et al. paper

Line 61: "were shown"; have been shown

Line 62: Please cite Tanos et al.

Line 73: "The latter two will be described in an accompanying paper." This needs to be rephrased.

Line 156 "Consistent with its function in ciliogenesis, we observed enhanced centriolar

localization of IFT88, which requires" IFT88 is not shown in Figure 1 or in supplementary figure 4C.

Figure 1: Could the Cep83 inner ring localization be shown in the absence of serum? Could this figure include INPP5E?

Line 164 "The previous study". Please rephrase, perhaps say: "previous work"

5. The hierarchy of recruitment of distal appendage proteins was published by Tanos et al. (PMID: 23348840). These results have been validated by many others. Originally, Tanos et al. used U2OS and RPE-1 cells and depleted the different proteins using siRNA to confirm this hierarchy. Please note that the updated hierarchy model has not been validated. As it is, I would suggest that Figure 2N would be updated or removed, as it does not reflect currently validated results.

6. These results shown need to be compared side by side with the results from siRNA and knockout cell lines. The authors state: "Notably, the original epistasis pathway using siRNA knockdowns for loss of function may fail to identify some strong requirements if limited amounts of protein are sufficient for pathway function." A risk of using knockout cell lines is whether genetic compensation might occur. Using a validated knock-in degron system could be an alternative that could give experimental support to these conclusions. To draw the conclusions observed in this paper, experiments similar to was has been done before, in order to provide valid comparisons should be carried out.

7. The results shown need to be validated using rescue experiments.

8. The quantification of protein localization has been carried out in an automated manner, focusing on a subdistal appendage protein "Cep170". In most papers examining the centriolar localization of proteins, careful attention needs to be paid to whether the image has been focused on the centriole. It is not clear how this might have been done in an automated manner. It would have been more reasonable to do the mask with a more broad centrosomal marker. It is not clear how the experiment was analyzed and normalized. Why does it say "when" 2 experiments were performed? It is not clear whether these results are representative of three independent experiments, two, or less. It is not clear whether a total of 100 cells were tested or 100 cells per repeat. Although the manual analysis was not performed, the data seems curated and it is not clear how this quantification was carried out.

9. Although this has not been published, we have examined CEP83 localization in SCLT1 knockout cells and we did not see that CEP83 was depleted from centrioles, which highlights the need for proper assay validation.

10. For Figure 2, the figure legend is not clear, is this representative of 3 independent experiments? How is the localization of CEP83 in SCLT1 knockout cells growing in a steady state situation, in 10% FBS, this needs to be addressed, as the experiments from Tanos et al. were conducted in 10% FBS. The data presented is not sufficient to conclude that CEP83 is not being recruited to centrioles in the absence of SCLT1. How many cells were analyzed for this experiment? How many cell lines were used? Representative images are not shown.

10. For the ANKRD26 (and for all experiments shown), no rescue experiments are shown that might validate the results.

11. The positive feedback loop by TTBK2 is not novel and was previously observed by Lo et al. (PMID: 31455668) and by Bernatik et al. (PMID: 32129703), this needs to be clearly stated in the text. Bernatik et al. showed that without Cep164, TTBK2 can't be recruited to centrioles. The model drawn in Figure 2N is confusing and misleading, as it is not supported by the data presented. I suggest the authors remove it.

12. For the FBF1 knockdown cells, it is mentioned that the FBF1 antibody does not work well for western blot. It has been shown by several papers that the FBF1 antibody works perfectly well (PMID: 23348840; PMID: 32366837; PMID: 33561696).

13. To examine the distal appendage hierarchy, they used an automated analysis of centrosomal localization. It is not clear how this was quantified and pictures are not shown. They used CEP170, a marker for subdistal appendages, to define a mask around centrioles. It is not clear how the experiment was analyzed and normalized. The techniques used can't be compared with experiments carried out by the community as those used different methods including storm and other super-resolution techniques less prone to artifacts and correlated electron microscopy. Rescue experiments should be included.

14. For the different positive feedback loops, representative images need to be shown. It is not clear how ANKRD26 can affect CEP83 localization. It is important to note that the localization of CEP83 was shown not to be affected in ANKRD26 knockout cells in a paper by the Loncarek group (PMID: 30824690).

15. Have these experiments been carried out in additional cell lines? Not only RPE-1? Careful analysis is required to reconcile all the previous models using comparable techniques and these inconsistencies need to be clearly mentioned in the text.

16. Line 224 (Kurtulmus et al., 2018), we did not see the effect of LRRC45 on FBF1 localization (Figure 2E). The authors claim that this is due to siRNA vs knockout. It is not clear that whenever possible, the explanation is siRNA vs knockout.

17. Could you test the localization of these proteins with and without 10% serum?

18. Line 257 the statement "Albinism likely reflects problems in melanosome transport." Seems disconnected, and needs to be rephrased.

19. Figure 3, it is not clear in this figure where the CEP83 cells are. The description in line 263 is vague, it says: "arrowhead in the bottom of Figure 3" This is not clear.

20. Where it says, line 275: "Note that the percentage of RAB34 and MYO5A double positive centrioles was highest in the confluent cells grown with serum and was decreased upon serum starvation (Figure 3B)." why would this be? Is this all centriolar localization or pericentriolar? This would need to be addressed.

21. In figure 3, however, Rab34 ciliary vesicle seems small puncta. I would expect a larger ciliary vesicle to be recruited.

22. Figure 4: It is not clear why the Rab34 data is included. It is possible that the results are inconsistent with previous studies (Ganga, 2021, Stuck 2021) because they did the experiments for 48 hours.

It is consensus to carry out ciliogenesis studies for 48 hours. Minimally, I would expect the authors to carry out a time course. What about the recruitment of TTBK2?

23. Figure 4. It is not clear how the Rab34 recruitment was increased in FOP knockout cells. It is not a slight increase, it is a very big difference. How was this experiment quantified? Could they do a time course? Could representative images be shown? The model in the figure is confusing. Could other cell lines be used, U2OS, HeLa, and IMCD3? Could this be an artifact?

24. Line 404, Tanos et al. also showed superresolution of novel distal appendage proteins CEP83, CEP89, SCLT1, and FBF1. Why is this not being recognized here?

25. Line 439, it was Tanos et al. that showed that distal appendages, and, in particular, CEP83 were required for CP110 removal. At the time the Goetz paper came out, TTBK2 had not yet been identified as a distal appendage protein. So, Tanos et al. first showed that CEP83 (and distal appendages) were required for CP110 removal.

26. Line 474, what does the author mean by "other distal appendages"?

27. Figure 1—figure supplement 1, the western blot for CEP83 is not clear, the molecular weight of the CEP83 band is at almost 100 KD in the left panel in 1B. The western blot does not seem specific.

28. FBF1 knockout needs to be validated by western blot.

29. Typos in the figures, Figure 6, should say "structural".

30. Figure 4—figure supplement 2 Characterization of IFT52 depleted cell, should say "cells".

31. Figure 5—figure supplement 1, should say "length".

*Reviewer #3 (Recommendations for the authors):*

1) The authors showed that the localization of the outer ring of CEP83 was significantly enhanced following serum starvation (Figure 1D). The authors should consider checking if the inner ring of the CEP83 signal also changes upon serum starvation. This may provide evidence if serum starvation could change the confirmation of the distal appendage.

2) In Figure 3, the authors showed that the percentage of RAB34 and MYO5A double positive centrioles was highest in the confluent cells grown with serum and was decreased upon serum starvation. Adding control analysis of RAB34 positive centrioles in non-confluent cells growing in serum could be used to evaluate the strength of promotion of ciliary vesicle formation upon serum starvation.

3) The application of single-molecule 3D super-resolution microscopy introduces an important new tool for investigating cilia membrane relationships. From the work presented only a limited number of cells were analyzed preventing quantitative analysis of vesicle docking. Including additional data from this microscope, modality could enhance conclusions that can be drawn in particular comparing cells that not undergoing ciliogenesis.

4) The authors set out to characterize the structural organization and function of the previously known and newly described complement of distal appendage protein. In agreement with previous studies, this work confirms that the four integral components of the distal appendage CEP83-SCLT1-CEP164-TTBK2 are not only important for structural integrity but also critical for ciliogenesis. Without either of these proteins, ciliogenesis initiation is strongly inhibited except for CEP19 recruitment in the CEP83 knockout condition. Their work also suggests additional distal appendage protein complexes have specific ciliogenesis functions. The authors predict that a very small amount of centriolar CEP164-TTBK2 may be sufficient to bring CEP19 near the distal appendage. Given the fact that CEP83 knockout strongly blocks distal appendage assembly, it doesn't rule out the possibility that CEP19 may be recruited to the other compartment of the mother centriole such as the sub-distal appendages in the absence of distal appendages. Showing the precise localization of CEP19 in the absence of CEP83 using super-resolution microscopy could strengthen their model.

5) The authors' work suggests differences in findings from knockout cells for RAB34 and MYO5A compared to previously published work. Rescue studies in knockout lines should be considered in cases where partial phenotypes were observed to ensure effects are specific, for example in the case of CP110 removal. Another important distinction in this work is the observation MYO5A knockout cells reported in this manuscript develop cilia which differs from Wu et al. (Nature Cell Biology, 2018). An explanation for this finding should be addressed in the text.

6) Additional editing of the text is suggested to improve the manuscript.

---

## [Author Response]

Essential revisions:Reviewer #1 (Recommendations for the authors):A major new finding of the work is that Cep83 is an extended protein that provides the backbone for much of the distal appendage. This conclusion is drawn from the localization of two antibodies that show extensive cross-reactivity to a ladder of cellular proteins. While I appreciate that the signal from these antibodies is reduced at the centrosome in the Cep83 knockout lines, alternative explanations could be made. To strengthen this conclusion, the authors should rescue the knockout cell line with constructs tagged at various positions. Representative images used for the quantification in Figure 2 should be shown.

Although centriolar CEP83 fluorescent signals that were probed with the two different antibodies were undetectable in CEP83 knockouts cells, we agree that we cannot exclude that possibility that these antibodies are specifically detecting distal appendage proteins that are not CEP83 but the downstream of it. To exclude this possibility and support our hypothesis, we tested the localization of N-terminally GFP tagged CEP83, and C-terminally GFP-tagged CEP83. Consistent with the inner-ring localization seen using the antibody detecting the CEP83 C-terminus (Figure 1—figure supplement 2), C-terminally GFP-tagged CEP83 (CEP83-GFP) displayed an inner-ring structure with the diameter of 339.7±18.4 nm (new Figure 1—figure supplement 4A-B). An N-terminally GFP tagged CEP83 was located at the similar position to CEP164 with the diameter comparable to CEP164 (new Figure 1—figure supplement 4C-D). The N-terminally tagged GFP-CEP83 ring was smaller than the outermost ring of endogenous CEP83, possibly because the difference between N-terminus and the region detected by antibody (somewhere between a.a. 226-568 of the CEP83 isoform 2). We observed the same trend when we detected the tagged proteins with native GFP fluorescent signal instead of antibody, although the difference between GFP-CEP83 and CEP83-GFP was somewhat less pronounced possibly due to lower fluorescent signal, which tends to flatten differences on the low end (new Figure 1—figure supplement 4E-H). These results strengthen our first hypothesis that CEP83 forms an extended structure that serves as an important backbone of the distal appendage.

We hesitated to show representative images in the initial submission, as one picture cannot reflect the heterogeneity of the centriolar signal. Nevertheless, we understand that the representative images are convenient for readers, and decided to have a figure that shows representative images of the cells analyzed in Figure 2 (new Figure 2—figure supplement 1). To mitigate the problem of the representative images, we uploaded all the raw image data that were analyzed in this paper to BioImage Archive (accession: S-BIAD1215; DOI: 10.6019/S-BIAD1215) and are publicly available for consideration by readers.

The model presented in figure 6 appears to be more of a visual abstract rather than a diagram that would help to summarize the findings of the paper. Even after reading the paper and studying the figure legend, I have little understanding of what figure 6 is trying to show. This paper would have more impact if the figure showed the main points of the document. That being diagrams illustrating how the proteins are organized with respect to each other and another of the hierarchy of assembly.

Based on the comment, we updated the model to better summarize the finding of the paper. We believe the new figure better conveys what we showed in this paper (please see the new Figure 6). We appreciate the reviewer for this important comment.

Reviewer #2 (Recommendations for the authors):There are a number of points in this paper.1. Throughout the paper, there is a lack of recognition of the Tanos et al. paper, which initially described a number of points discussed in this paper. Although Tanos et al. is cited, the work not only identified four novel distal appendage proteins but also molecularly showed how distal appendages mediated centriole to membrane docking through the recruitment of TTBK2 and the removal of CP110 and the recruitment of IFT molecules. In addition, her careful EM analysis of polarized IMCD3 cells using serial sections transversally cut along an apicobasal axis, and TEM clearly showed that, in the absence of CEP83, centrioles did not dock to the plasma membrane. I assume that this is an unintentional omission and I have pointed out specific points below.

Our current paper was clearly built upon previous studies, which established our current understanding of the distal appendages. These include but are not limited to: (1) the paper from Eric Nigg’s group (PMID:17954613), which discovered the first distal appendage protein, CEP164; (2) the paper from Michel Bornens’ group (PMID: 21976302), which discovered CEP89 as the second distal appendage protein and performed the first STORM experiment on the distal appendage protein; (3) the paper from Gislene Pereira’s group (PMID: 23253480), which showed that the distal appendage protein, CEP164, is important for IFT recruitment and ciliary vesicle capture; (4) the Tanos et al. paper (PMID: 23348840), which discovered the three novel distal appendage proteins and established the first hierarchy of the known distal appendage proteins; and (5) recent super-resolution imaging papers (PMID: 29789620)(PMID: 30824690), which determined the precise location of the distal appendage proteins. We, of course, recognize, value and greatly appreciate all these papers, and had no intention of omitting any of those papers. The Tanos et al. paper was cited eleven times in the initial manuscript and is the most cited paper among all the references, so we believe we clearly emphasized her paper. We assume the reviewer felt the lack of recognition of the paper possibly because of our criteria of referencing, where we limit the references to the most relevant papers for the statement, rather than adding all papers that did something related to the statement.

2. To map the localization of centriole distal appendage proteins, the authors use 3D-SIM. Several studies have used STORM, correlated EM, and expansion microscopy to look at the structure and localization of distal appendages. This includes Yang et al. (PMID: 29789620), Bowler et al. (PMID: 30824690), and others. It, therefore, does not seem conclusive to use an inferior technique that is not properly validated by correlated EM. Controls with previously used techniques have not been carried out. Could additional cell lines be used to test these antibodies? Antibodies for centriolar proteins have been shown to frequently result in unspecific centriolar signals. Could these new antibodies be tested for immunostaining in knockout cell lines to see if the centriolar signal is specific?

The lateral resolution of 3D-SIM is indeed inferior to STORM; however, each super-resolution technique has its own merits and problems, and 3D-SIM has several advantages over other super-resolution techniques. One notable advantage is its flexibility of fluorophore selection, which allows us to perform the multi-color imaging to locate target proteins relative to multiple centriolar markers, which we use in the paper. In contrast, the success of STORM experiments highly relies on blinking status of the dye, and multi-color STORM imaging is always challenging, as second dye often has inferior blinking property compared with the gold standard, AlexaFluor647/CF647.

Importantly, we recapitulated many of the previous finding, including the distal appendage matrix localization of FBF1 (PMID: 29789620), and relative position of the previously known distal appendage proteins (please compare Figure 1B with the previously established architecture (PMID: 29789620)).

All the antibodies used in this paper were rigorously tested for both immunostaining and immunoblot. We confirmed that all the antibodies specifically recognize the respective proteins at least at the centriole (Figure 2).

3. Most of the DAP localization data shown is not new. However, three novel distal appendage proteins are described. Regarding the CEP83 localization, a novel localization in the outer rings of DAPs is found. Additional techniques should be used to confirm this. Furthermore, a number of antibodies for centriolar proteins show unspecific signals at centrioles. Has this antibody staining been tested in CEP83 knockout cells? A western blot is shown for the knockout cells but not for immunostaining. In "Figure 1—figure supplement 2 Characterization of the two CEP83 antibodies", here, the western blot is showing a band that seems to run at more than 90 KDa, the band does not seem to correspond to CEP83 and seems very unspecific. Has this localization been validated using EM? 3D-SIM has also been shown to generate artifacts, but this has not been taken into account.

We confirmed that the centriolar signal that was detected by either of the two CEP83 antibodies was undetectable in CEP83 knockout cells (please see Figure 2B and 2C). For the sake of convenience, we also included the representative images of the data analyzed in Figure 2 in the revised manuscript (new Figure 2-fgure supplement 1). We also uploaded all the raw images in the BioImage Archive (accession: S-BIAD1215; DOI: 10.6019/S-BIAD1215).

Further, we tested our hypothesis for the extended CEP83 structure by tagging the protein with GFP at either N-terminus or the C-terminus as we explained in the response to the comment 1 of the reviewer #1 (please see above). This experiment has its own limitation (e.g., the exact motifs predominantly recognized by the antibodies are unknown, N/C-terminally GFP tagged CEP83 does not necessarily mimic the motif recognized by the antibodies), but generally agree with the idea that CEP83 forms an extended structure that may span from inner to the outer region of the distal appendage.

Regarding the molecular weight of CEP83, estimated molecular weight using immunoblot is often different from the calculated molecular weight, as the migration speed on a protein gel is affected by numerous factors, including amino acid composition and efficacy of SDS binding, and post translational modification.

Nevertheless, we rechecked the figure (Figure 1—figure supplement 2) and noticed that the molecular weight for the outer ring was slightly displaced. As you can see in the original image “Source Data- the original files of the full raw unedited immunoblot”, which was included in the initial submission, the two membranes were slightly displaced when the image was taken. When we labeled the molecular weight during the manuscript preparation, we accidentally copied the molecular weight marker of the right panel (inner ring) and pasted them to the left panel, which caused the displacement of the molecular weight marker in the left panel. The figure was now updated (new Figure 1-fgure supplement 2B). The accuracy of this change can be confirmed with the original image (please see “Source Data- the original files of the full raw unedited immunoblot”). We apologize for this mistake and appreciate the reviewer for pointing this out.

4. Regarding this, please see the points below, line by line:Line 31 "Functional assay revealed that CEP89 selectively functions in RAB34+ ciliary vesicle.Recruitment", should say "functional assays". Please correct the language throughout.

Thank you very much for pointing this out.

Line 50 "Tsou, 2016?" what reference is this? Is this in the references section?

This reference is:

Mazo, G., Soplop, N., Wang, W. J., Uryu, K., and Tsou, M. F. (2016). Spatial Control of Primary Ciliogenesis by Subdistal Appendages Alters Sensation-Associated Properties of Cilia. Dev Cell, 39(4), 424-437. doi:10.1016/j.devcel.2016.10.006

Yes, it was included in the reference section.

Line 54 says: "2 recruitment of intra-flagellar transport (IFT) protein complexes (Cajanek and Nigg, 2014; Goetz, Liem, and Anderson, 2012; Schmidt et al., 2012)"Here, Tanos et al. should be cited (PMID: 23348840), as this was also shown in the Tanos et al. paper

The reference went to the papers that reported the importance of the distal appendage proteins in recruitment of the IFT proteins for the first time. To include the nuance that several IFT-B complex proteins as well as IFT-A complex protein were shown to be dependent on the distal appendage, we cited several papers, each of which showed a different IFT component. We cited [Schmidt et al., 2012] for IFT88, [Goetz, Liem, and Anderson, 2012] for IFT140 (IFT-A), and [Cajanek and Nigg, 2014] for IFT81.

Line 61: "were shown"; have been shown

Corrected.

Line 62: Please cite Tanos et al.

This citation went to the paper that described the distal appendage protein for the first time. The first paper that showed the localization of CEP89 was [Sillibourne et al., 2011] (PMID: 21976302), so we cited this paper together with the follow-up from the same authors with the functional characterization [Sillibourne et al., 2013] (PMID: 23789104).

Line 73: "The latter two will be described in an accompanying paper." This needs to be rephrased.

The final paper will have the citation to the “accompanying paper”, which we believe help readers to understand this statement correctly.

Line 156 "Consistent with its function in ciliogenesis, we observed enhanced centriolarlocalization of IFT88, which requires" IFT88 is not shown in Figure 1 or in supplementary figure 4C.

It was shown in the Figure 1—figure supplement 4C. If the reviewer was asking for a representative image, we uploaded all the raw images used for quantification of the centriolar signal to the BioImage Archive (accession: S-BIAD1215; DOI: 10.6019/S-BIAD1215).

Figure 1: Could the Cep83 inner ring localization be shown in the absence of serum? Could this figure include INPP5E?

We could not include the inner ring localization of CEP83 in the absence of serum in either Figure 1D or Figure 2B, because the antibody that recognizes inner ring of CEP83 showed strong signal surrounding the centriole (likely the Golgi apparatus) only in the absence of serum. We confirmed that the Golgi-like localization is likely non-specific, as this staining persists in the CEP83 knockout cells. This non-centriolar signal was problematic especially with our workflow of automated fluorescent signal measurement.

We could not include INPP5E localization in Figure 1D, as the centriolar localization of INPP5E is highly variable (some cells have INPP5E at the centriole and the other do not). Also, strong ciliary localization of INPP5E prevents us from accurately measuring centriolar signal. Depending on the orientation, centriolar signal is highly overlapping with ciliary signal.

Line 164 "The previous study". Please rephrase, perhaps say: "previous work"

Changed to “previous work”.

5. The hierarchy of recruitment of distal appendage proteins was published by Tanos et al. (PMID: 23348840). These results have been validated by many others. Originally, Tanos et al. used U2OS and RPE-1 cells and depleted the different proteins using siRNA to confirm this hierarchy. Please note that the updated hierarchy model has not been validated. As it is, I would suggest that Figure 2N would be updated or removed, as it does not reflect currently validated results.

Figure 2N shows a model based on our study. We believe this model helps readers to better understand our study. However, we thought having CEP83 and SCLT1 at the same hierarchy (the original Figure 2N) may cause a confusion, as the inner ring CEP83 was much less affected in SCLT1 knockout cells (Figure 2B) while the outer ring CEP83 was almost undetectable (Figure 2C) in SCLT1 knockout cells. Therefore, we decided to update our model (please see the updated Figure 2N).

6. These results shown need to be compared side by side with the results from siRNA and knockout cell lines. The authors state: "Notably, the original epistasis pathway using siRNA knockdowns for loss of function may fail to identify some strong requirements if limited amounts of protein are sufficient for pathway function." A risk of using knockout cell lines is whether genetic compensation might occur. Using a validated knock-in degron system could be an alternative that could give experimental support to these conclusions. To draw the conclusions observed in this paper, experiments similar to was has been done before, in order to provide valid comparisons should be carried out.

Each technique has its own advantage(s) and disadvantage(s). We understand the potential risk of analyzing knockout cells (e.g., compensation). A degron system may be powerful in terms of temporal regulation, but the reduction of the protein level may not be sufficient to completely shut off function of the proteins. The issue of genetic compensation is well-understood and should not confuse readers. We provide a short statement to address this issue. In the discussion, we wrote “Using the CRISPR-Cas9 system, we created a new hierarchical model of the distal appendage proteins, which is mostly consistent with the previously established model but has a number of differences. Most of the discrepancies likely come from the differences between siRNA and CRISPR techniques. While CRISPR knockout lines may show genetic compensation, the consistency of our results with the previous CRISPR and siRNA results, in some cases showing only quantitative differences, do not suggest that genetic compensation is a major problem.”

Importantly, our knockout cells reproduced most of the previously published results with some differences. We believe that it is fair to state that stronger phenotypes that we observed may likely come from complete absence of the proteins in the knockout cells. For example, our CEP83 knockout cells, SCLT1 knockout cells, and CEP164 knockout cells had slightly stronger ciliation defects (Figure 5A-B) than what were observed in the Tanos et al. paper (PMID: 23348840). The very similar stronger ciliation defects were observed in another paper that analyzed CEP164 knockout cells and SCLT1 knockout cells (please see the Figure EV1 of PMID: 33350486).

Nevertheless, we believe the different techniques provide generally interpretable views of the importance of the distal appendage proteins. Our intent was not to analyze the differences between siRNA and CRISPR techniques, but only to provide a self-consistent data set analyzing the distal appendage proteins.

7. The results shown need to be validated using rescue experiments.

We strongly agree that perfect experiments that confirm the results are rescue experiments. We wish we could perform rescue experiments for all the knockout cells, however, it could take additional 2-3 years to complete the rescue experiments especially because we had >15 knockout cells in this study. Nevertheless, we understand that knockout cells may have off target effect and mitigated this problem using an alternative way. To reduce the risk of off-target effect, we generated multiple single cell clones from each knockout cells, and the same number of the single cell clones were mixed to generate pooled single cell clone knockout cells. The pooled single cell clones were used in all the experiments shown in this paper, and the number of clones used to generate each knockout line was reported in the “Source Data_List of cell lines used in this paper”. With this strategy, each knockout clone should have similar on-target effect, but likely have different off-target effects. Therefore, the phenotypes observed in majority of the cells are likely attributable to the depletion of the target genes.

8. The quantification of protein localization has been carried out in an automated manner, focusing on a subdistal appendage protein "Cep170". In most papers examining the centriolar localization of proteins, careful attention needs to be paid to whether the image has been focused on the centriole. It is not clear how this might have been done in an automated manner. It would have been more reasonable to do the mask with a more broad centrosomal marker. It is not clear how the experiment was analyzed and normalized. Why does it say "when" 2 experiments were performed? It is not clear whether these results are representative of three independent experiments, two, or less. It is not clear whether a total of 100 cells were tested or 100 cells per repeat. Although the manual analysis was not performed, the data seems curated and it is not clear how this quantification was carried out.

We are afraid that we may not have understood the reviewer’s questions correctly. Quantification was carried out in an automated manner, but each image was manually captured by adjusting the focus to CEP170 without looking at a channel of the protein of interest (POI) to avoid selecting specific area based on the signal intensity of POI. Distal appendage proteins are typically in the same focal plane as CEP170, because the subdistal appendage protein is in proximity to distal appendage proteins. All the images analyzed were single-plane pictures (not z-stacks). We removed the unfocused centrioles by removing outliers using the ROUT method. The exact workflow used for the quantification was described in the “Fluorescence intensity measurements” of the Materials and methods, and the macro used for the analysis was provided in “Source Data- Macro for measuring fluorescent intensity of centrosomal proteins”. In the typical analyses, 50-100 centrioles per cell line were analyzed in each experiment. Exact numbers of the centrioles analyzed, and the raw signal intensity can be found in the “Source Data” of the corresponding figures. The raw images can be found through the BioImage Archive (accession: S-BIAD1215; DOI: 10.6019/S-BIAD1215). In majority of the case, including CEP164, CEP89, SCLT1, CEP83, and TTBK2, we performed more than three independent experiments (four to six) to confirm our observation. In the case of ANKRD26, FBF1, LRRC45, and KIZ, we performed two independent experiments.

We do not think any of our images are selected (if that is the meaning of “curated”) to show anything but a true representation of what we see in the data.

9. Although this has not been published, we have examined CEP83 localization in SCLT1 knockout cells and we did not see that CEP83 was depleted from centrioles, which highlights the need for proper assay validation.

We highly recommend quantifying the fluorescent signal, when inner ring of CEP83 is analyzed. The depletion of signal from outer ring of CEP83 should be obvious in SCLT1 knockout cells. Please use the CEP83 antibody from Proteintech Group (cat#26013-1-AP) to detect outer ring of CEP83. Unpublished studies are hard to track and cannot be easily cited, so we cannot directly address the reviewer’s note, but we are confident in our analysis.

10. For Figure 2, the figure legend is not clear, is this representative of 3 independent experiments? How is the localization of CEP83 in SCLT1 knockout cells growing in a steady state situation, in 10% FBS, this needs to be addressed, as the experiments from Tanos et al. were conducted in 10% FBS. The data presented is not sufficient to conclude that CEP83 is not being recruited to centrioles in the absence of SCLT1. How many cells were analyzed for this experiment? How many cell lines were used? Representative images are not shown.

The figures shown in Figure 2 are representative quantification data of at least two independent experiments. In majority of the case, including CEP164, CEP89, SCLT1, CEP83, and TTBK2, we performed more than three independent experiments (four to six) to confirm our observation. In the case of ANKRD26, FBF1, LRRC45, and KIZ, we performed two independent experiments.

We noticed that our figure legend regarding CEP83 ring (Figure 2B) was incorrect in the initial submission. As the reviewer mentioned, this was indeed performed in the cells with 10% FBS, as this antibody showed strong peri-centriolar staining in the serum starved cells, which prevented us to perform accurate calculation. Although this condition was properly written in the “Figure 2B-Source Data” in the initial submission, we failed to indicate the proper cell culture condition in the legend of Figure 2B. We corrected this in the revised manuscript. Thank you very much for pointing this out.

The exact number of cells analyzed in each line can be found in the “Figure 2B-Source Data”. Each knockout line is a mixture of multiple single cell clones. Exact number of the single cell clones included in each line is listed in “Source Data_List of cell lines used in this paper”.

11. For the ANKRD26 (and for all experiments shown), no rescue experiments are shown that might validate the results.

We answered this question in “Response to the comment 7 of the reviewer #2”.

12. The positive feedback loop by TTBK2 is not novel and was previously observed by Lo et al. (PMID: 31455668) and by Bernatik et al. (PMID: 32129703), this needs to be clearly stated in the text. Bernatik et al. showed that without Cep164, TTBK2 can't be recruited to centrioles. The model drawn in Figure 2N is confusing and misleading, as it is not supported by the data presented. I suggest the authors remove it.

We showed that TTBK2 is required for structural integrity of distal appendage by stabilizing multiple distal appendage proteins, including outer region of CEP83, CEP164, ANKRD26, FBF1, and NCS1. To our knowledge, this has not been clearly shown in previous studies. Lo et al. (PMID: 31455668) showed that CEP83 is phosphorylated by TTBK2, and the phosphorylation of CEP83 promotes cilium formation without affecting localization of other distal appendage proteins. Similarly, Bernatik et al. (PMID: 32129703) paper showed that TTBK2 phosphorylates CEP164, but it was not shown that CEP164 localization was affected by depletion of TTBK2. The model shown in the Figure 2N was generated simply by the data presented in Figure 2.

13. For the FBF1 knockdown cells, it is mentioned that the FBF1 antibody does not work well for western blot. It has been shown by several papers that the FBF1 antibody works perfectly well (PMID: 23348840; PMID: 32366837; PMID: 33561696).

In the PMID: 23348840, we could not find immunoblot data showing FBF1. In the PMID: 32366837 paper, FBF1 antibody (11531-1-AP, Proteintech) was used to detect FBF1 in the ANKRD26/TALPID3 immuno-precipitates. They detected the band around 130 kDa. The same antibody was used in PMID: 33561696, and the authors detected the band around 100 kDa. Using the same antibody, we observed the strong band at -70kDa and faint band at -160 kDa with several additional bands (please see Author response image 1). None of them were significantly diminished in FBF1 knockout cells (although the band at 160kDa may be slightly decreased). The same antibody used for immunostaining detected centrosomal signal, which is significantly reduced in all the knockout clones (clone#1-9) of FBF1 knockout cells. Therefore, we concluded that this does not work well for immunoblot.

**Author response image 1. sa2fig1:** 

14. To examine the distal appendage hierarchy, they used an automated analysis of centrosomal localization. It is not clear how this was quantified and pictures are not shown. They used CEP170, a marker for subdistal appendages, to define a mask around centrioles. It is not clear how the experiment was analyzed and normalized. The techniques used can't be compared with experiments carried out by the community as those used different methods including storm and other super-resolution techniques less prone to artifacts and correlated electron microscopy. Rescue experiments should be included.

We believe that we addressed this question in “Response to the comment 8 of the reviewer #2”

15. For the different positive feedback loops, representative images need to be shown. It is not clear how ANKRD26 can affect CEP83 localization. It is important to note that the localization of CEP83 was shown not to be affected in ANKRD26 knockout cells in a paper by the Loncarek group (PMID: 30824690).

The Loncarek group’s paper used the CEP83 (HPA038161) antibody, which shows inner-ring structure (Figure 1A), and could not see the reduction of the centriolar CEP83 signal in ANKRD26 knockout cells. We observed the same result using the same antibody (please see Figure 2B). However, when we used the CEP83 antibody (26013-1-AP), which detects outer-ring structure (Figure 1A), we were unable to detect CEP83 signal in ANKRD26 knockouts. We think ANKRD26 stabilizes CEP83 only at the outer region of the distal appendage.

16. Have these experiments been carried out in additional cell lines? Not only RPE-1? Careful analysis is required to reconcile all the previous models using comparable techniques and these inconsistencies need to be clearly mentioned in the text.

These experiments were only carried out using RPE-hTERT cells. We believe that we performed most of experiments with optimal techniques and reasonably discussed the inconsistencies in the text.

17. Line 224 (Kurtulmus et al., 2018), we did not see the effect of LRRC45 on FBF1 localization (Figure 2E). The authors claim that this is due to siRNA vs knockout. It is not clear that whenever possible, the explanation is siRNA vs knockout.

We believe the differences between siRNA and knockout is the most reasonable explanation for the discrepancy.

The milder phenotype that we observed may be genetic compensation, or the phenotype may be due to an off-target effect.

18. Could you test the localization of these proteins with and without 10% serum?

The localization of the most of proteins described in this paper was tested both in the presence or the absence of serum. Please see Figure 1D. If the reviewer is asking for raw images, all the images were uploaded to BioImage Archive (accession: S-BIAD1215; DOI: 10.6019/S-BIAD1215)

19. Line 257 the statement "Albinism likely reflects problems in melanosome transport." Seems disconnected, and needs to be rephrased.

We rephrased it to “The albinism is likely due to the defect in melanosome transport rather than dysfunction of the cilium.”

20. Figure 3, it is not clear in this figure where the CEP83 cells are. The description in line 263 is vague, it says: "arrowhead in the bottom of Figure 3" This is not clear.

This statement refers Figure 3—figure supplement 1A, but not Figure 3.

I understand that this format is a bit confusing, but this format is commonly used in *eLife*.

21. Where it says, line 275: "Note that the percentage of RAB34 and MYO5A double positive centrioles was highest in the confluent cells grown with serum and was decreased upon serum starvation (Figure 3B)." why would this be? Is this all centriolar localization or pericentriolar? This would need to be addressed.

Since RAB34 does not show pericentriolar localization, RAB34/MYO5A double positive dots are only found at centriole. Previous studies reported the same observation [PMID: 29335527][PMID: 33989524][PMID: 33989527]. It is likely because RAB34/MYO5A is required for the early step of the cilium formation, but not for the maintenance.

22. In figure 3, however, Rab34 ciliary vesicle seems small puncta. I would expect a larger ciliary vesicle to be recruited.

In the 1962 paper, Dr. Sorokin named a vesicle that covers the entire width of the centriole as a primary ciliary vesicle (PMID: 13978319). In the follow-up study in 1968, he observed small vesicles, which may attach to one of the nine blades of the distal appendage (PMID: 5661997). He did not clearly name these smaller vesicles, but when he explained ciliogenesis of multi-ciliated cells, he stated " it is possible to observe several small vesicles at the distal end, or a single vesicle covering that end, of a mature basal body (Figures 51, 53-55); and this may be interpreted to be either a rudimentary ciliary sheath (the primary ciliary vesicle) or an invagination from the plasma membrane." So, the small vesicle can be considered as a ciliary vesicle from this paper. Westlake lab named this smaller vesicle as a distal appendage vesicle, when they described the role of EHD1, which converts smaller vesicles to a larger ciliary vesicle (PMID: 25686250). Since RAB34 is recruited to the centriole earlier than EHD1, RAB34 may be classified as a distal appendage vesicle protein. However, we hesitate to use the word “distal appendage vesicle” from several reasons. First, we think that the current definition of the ciliary vesicle/distal appendage vesicle is not very clear. When we analyzed serial section electron micrographs in RPE-hTERT cells, we found a number of different-sized vesicles attached to the mother centrioles. Some vesicles occupied only several of the nine blades of the distal appendage. Please see electron micrographs of the Figure 3D of the accompanying paper. Some middle-sized vesicles formed tubule-like structure, reminiscent of the MC membrane tubule described by the Westlake group. Some vesicles are large even though it attaches to only one blade of the distal appendage. Even one centriole can have multiple of those different sized vesicles. We were not sure how we should call those vesicles. Second, when we analyze the RAB34 positive vesicles using wide-field microscopy, we cannot distinguish small and large vesicles because of the diffraction limit. Therefore, we prefer to use the word “ciliary vesicle” to all the vesicles that are attached to the mother centriole without shaft.

23. Figure 4: It is not clear why the Rab34 data is included. It is possible that the results are inconsistent with previous studies (Ganga, 2021, Stuck 2021) because they did the experiments for 48 hours.It is consensus to carry out ciliogenesis studies for 48 hours. Minimally, I would expect the authors to carry out a time course. What about the recruitment of TTBK2?

Distal appendages regulate at least four steps required for cilium formation: IFT/CEP19 recruitment, ciliary vesicle recruitment, and CP110 removal. We wanted to determine whether distal appendage regulates these steps independently, or these steps are interconnected. To determine the relationship, we wanted to individually inhibit each step and tested whether the perturbation of one process affects the others. To this end, we depleted RAB34 to inhibit ciliary vesicle formation.

In terms of the assay time point after serum starvation, we highly recommend analyzing the cilium formation at or earlier than 24 hours after serum removal, or perform kinetic experiments. As shown in many previous papers, RPE1-hTERT cells complete cilium formation sometime closer to 24 hours rather than 48 hours (please see Figure 3C of PMID: 28625565, for example). If the cilium formation or other processes are assessed at 48 hours, we may miss kinetic defects.

TTBK2 recruitment was not affected by RAB34 depletion (our unpublished data)

24. Figure 4. It is not clear how the Rab34 recruitment was increased in FOP knockout cells. It is not a slight increase, it is a very big difference. How was this experiment quantified? Could they do a time course? Could representative images be shown? The model in the figure is confusing. Could other cell lines be used, U2OS, HeLa, and IMCD3? Could this be an artifact?

FOP plays an important role for CEP19 and IFT88 recruitment to the centriole. An intriguing possibility is that RAB34 positive ciliary vesicle is stuck at the centriole when the other processes do not proceed. However, this hypothesis conflicts with the data where neither depletion of IFT52 (Figure 4L), which abrogated localization of all the IFT-B complex proteins that we tested, nor CEP19 (not shown) affected the percentage of RAB34 positive centrioles. Another possibility is that RAB34 recruitment occurs at specific cell cycle stage, as depletion of FOP causes a growth defect, possibly via regulation of PPP2R3C-MAP3K1 module (PMID: 38617270). This is very interesting phenomenon, but substantial amount of work is needed to understand exactly why RAB34 recruitment is increased in FOP knockout cells.

25. Line 404, Tanos et al. also showed superresolution of novel distal appendage proteins CEP83, CEP89, SCLT1, and FBF1. Why is this not being recognized here?

These references are for the sentence “the detailed protein architecture has been visualized mostly by super-resolution microscopy”. We believe the two referenced papers are the best fit for this sentence, as they revealed the detailed architecture of the distal appendage.

26. Line 439, it was Tanos et al. that showed that distal appendages, and, in particular, CEP83 were required for CP110 removal. At the time the Goetz paper came out, TTBK2 had not yet been identified as a distal appendage protein. So, Tanos et al. first showed that CEP83 (and distal appendages) were required for CP110 removal.

These references went to the papers that first linked distal appendage proteins to each downstream process. Yes, we wondered if we should refer the Goetz paper (PMID: 23141541) or the Tanos paper (PMID: 23348840), since the Goetz paper did not classify TTBK2 as the distal appendage protein. We rethought about it and decided to add the Tanos paper as a reference here. We appreciate the comment.

27. Line 474, what does the author mean by "other distal appendages"?

We changed this to “other distal appendage proteins”. We appreciate the reviewer for pointing this out.

28. Figure 1—figure supplement 1, the western blot for CEP83 is not clear, the molecular weight of the CEP83 band is at almost 100 KD in the left panel in 1B. The western blot does not seem specific.

We addressed this in the “Response to the comment 3 of the reviewer #2”.

29. FBF1 knockout needs to be validated by western blot.

We addressed this above. Please see “Response to the comment 13 of the reviewer #2”.

30. Typos in the figures, Figure 6, should say "structural".

We updated the Figure 6.

31. Figure 4—figure supplement 2 Characterization of IFT52 depleted cell, should say "cells".

Corrected. Thank you very much for pointing this out.

32. Figure 5—figure supplement 1, should say "length".

Corrected. Thank you very much for pointing this out.

Reviewer #3 (Recommendations for the authors):1) The authors showed that the localization of the outer ring of CEP83 was significantly enhanced following serum starvation (Figure 1D). The authors should consider checking if the inner ring of the CEP83 signal also changes upon serum starvation. This may provide evidence if serum starvation could change the confirmation of the distal appendage.

We could not include the inner ring localization of CEP83 in the absence of serum in either Figure 1D or Figure 2B, because the antibody that recognizes inner ring of CEP83 showed strong signal surrounding the centriole (likely the Golgi apparatus) only in the absence of serum. We confirmed that the Golgi-like localization is likely non-specific, as this staining persists in the CEP83 knockout cells.

2) In Figure 3, the authors showed that the percentage of RAB34 and MYO5A double positive centrioles was highest in the confluent cells grown with serum and was decreased upon serum starvation. Adding control analysis of RAB34 positive centrioles in non-confluent cells growing in serum could be used to evaluate the strength of promotion of ciliary vesicle formation upon serum starvation.

We agree that it is important to show whether the centriolar localization is affected by confluency. To test this possibility, we analyzed the percentage of RAB34 positive centrioles in either cells grown to 40-50% confluency (subconfluent) or 100% confluency (confluent). As shown in the new Figure 3—figure supplement 2A, we did not observe any difference between the two conditions. This result confirmed that the RAB34 positive ciliary vesicles can attach to the mother centriole in cycling cells, and the number of the RAB34-positive centrioles decreases over time during the time course of serum starvation. We explained the new figure in the main text.

3) The application of single-molecule 3D super-resolution microscopy introduces an important new tool for investigating cilia membrane relationships. From the work presented only a limited number of cells were analyzed preventing quantitative analysis of vesicle docking. Including additional data from this microscope, modality could enhance conclusions that can be drawn in particular comparing cells that not undergoing ciliogenesis.

We agree that additional analyses will add valuable insights into organization of ciliary/ciliary vesicle membrane, and will pursue this in our future studies.

4) The authors set out to characterize the structural organization and function of the previously known and newly described complement of distal appendage protein. In agreement with previous studies, this work confirms that the four integral components of the distal appendage CEP83-SCLT1-CEP164-TTBK2 are not only important for structural integrity but also critical for ciliogenesis. Without either of these proteins, ciliogenesis initiation is strongly inhibited except for CEP19 recruitment in the CEP83 knockout condition. Their work also suggests additional distal appendage protein complexes have specific ciliogenesis functions. The authors predict that a very small amount of centriolar CEP164-TTBK2 may be sufficient to bring CEP19 near the distal appendage. Given the fact that CEP83 knockout strongly blocks distal appendage assembly, it doesn't rule out the possibility that CEP19 may be recruited to the other compartment of the mother centriole such as the sub-distal appendages in the absence of distal appendages. Showing the precise localization of CEP19 in the absence of CEP83 using super-resolution microscopy could strengthen their model.

We reported that CEP19 localizes to the position between subdistal appendage and distal appendage, and the localization requires FGFR1OP (or FOP) [PMID: 28625565]. In FGFR1OP knockout cells, CEP19 almost completely fails to localize to the centriole (Figure 4I). As we showed in Figure 5F, CEP19 centriolar localization is highly dependent on the CEP164-TTBK2 module. This indicates that CEP19 requires both CEP164-TTBK2 and FOP to properly localize to the centriole. An interesting possibility is that CEP164 may be required for FOP localization and vice versa. We tested this possibility. Quantification data from the immunostaining of CEP164 showed that FGFR1OP is not required for CEP164 localization (Author response image 2). While quantification and more careful analysis is needed, immunostaining image showed that there is no obvious difference in centriolar FGFR1OP intensity in control, CEP164 knockout, and CEP83 knockout cells (Author response image 2). Therefore, the difference in CEP19 intensity between CEP164 and CEP83 knockouts is independent of FGFR1OP regulation. We believe this further supports our hypothesis that the residual amount (5-10%, please see Figure 2A) of CEP164 may be sufficient for CEP19 localization in CEP83 knockout cells.

5) The authors' work suggests differences in findings from knockout cells for RAB34 and MYO5A compared to previously published work. Rescue studies in knockout lines should be considered in cases where partial phenotypes were observed to ensure effects are specific, for example in the case of CP110 removal. Another important distinction in this work is the observation MYO5A knockout cells reported in this manuscript develop cilia which differs from Wu et al. (Nature Cell Biology, 2018). An explanation for this finding should be addressed in the text.

We strongly agree that a perfect experiment that confirms the result is a rescue experiment. While we did not confirm that CP110 removal defect observed in RAB34 knockout cells are rescuable, we confirmed that cilium formation defect of our RAB34 knockout cells is rescuable (See Author response image 3). Since CP110 removal is important step for cilium formation, we believe the CP110 removal defect that we observed in our RAB34 knockout cells is rescuable.

Regarding the lack of ciliation defect in MYO5A knockouts, which differs from the previous report, Wu et al. (Nature Cell Biology, 2018) [PMID: 29335527], we added several sentences to address this discrepancy in the revised manuscript. We wrote “The absence of ciliation defect in MYO5A knockout cells differs from the previous observation {Wu, 2018 #54}, but is consistent with the fact that mutations in MYO5A gene cause Griscelli syndrome {Pastural, 1997 #52}, of which phenotypes are distinct from ciliopathies.”

**Author response image 3. sa2fig3:** 

6) Additional editing of the text is suggested to improve the manuscript.

We further edited the manuscript.